# Nonlocal electrical detection of reciprocal orbital Edelstein effect

Weiguang Gao [1,10], Liyang Liao [1,10], Hironari Isshiki [1], Nico Budai [1], Junyeon Kim[2,3], Hyun-Woo Lee [4,5], Kyung-Jin Lee [6], Dongwook Go [7,8], Yuriy Mokrousov [7,8], Shinji Miwa [1,9] & Yoshichika Otani [1,2,9] ✉

The orbital Edelstein effect and orbital Hall effect, where a charge current induces a nonequilibrium orbital angular momentum, offer a promising method for efficiently manipulating nanomagnets using light elements. Despite extensive research, understanding the Onsager's reciprocity of orbital transport remains elusive. In this study, we experimentally demonstrate the Onsager's reciprocity of orbital transport in an orbital Edelstein system by utilizing nonlocal measurements. This method enables the precise identification of the chemical potential generated by orbital accumulation, avoiding the limitations associated with local measurements. We observe that the direct and inverse orbital-charge conversion processes produce identical electric voltages, confirming Onsager's reciprocity in orbital transport. Additionally, we find that the orbital decay length, approximately 100 nm at room temperature, is independent of the Cu thickness and decreases with decreasing temperature, revealing a distinct contrast to the spin transport behavior. Our findings provide valuable insights into both the reciprocity of the charge-orbital interconversion and the nonlocal correlation of orbital degree of freedom, laying the ground for orbitronics devices with long-range interconnections.

Research on the dynamics and transport of different electronic degrees of freedom has driven advancements in physics, material science, and device engineering. Spin angular momenta (SAM) and their conversion, transport, and interaction with magnetization have been extensively studied in spintronics over the past decades. The spintronics community has utilized charge-to-spin conversion mechanisms, such as the well-known spin Hall effect[1,2] and spin Edelstein effect[3,4], to generate nonequilibrium accumulation. Recently, the emerging field of orbitronics, which focuses on orbital angular momenta (OAM)[5–8], offers a complementary direction to the traditional reliance on SAM. Similar to

charge-to-spin conversion, charge-to-orbital conversion occurs through the orbital Hall effect (OHE)[5–7,9,10] and orbital Edelstein effect (OEE)[11–14]. Nonequilibrium OAM can be electrically induced in various light element systems with negligible spin-orbit coupling (SOC), such as Cr[15–17], Ti[5,18], Mn[16,19], $Al_2O_3$/Ru interface[20], oxides/Cu interface[21,22], and naturally oxidized Cu[23–27], as revealed by optical[5], terahertz emission[9], torque[11,17,20–22,28], magnetoresistance[16,24,29] and magnon[30] measurements, demonstrating the high efficiency of orbital torque and the broad selection of orbital source materials. However, these studies relied on local measurements, where the regions for OAM generation, distribution, and conversion

[1]Institute for Solid State Physics, The University of Tokyo, Kashiwa, Chiba, Japan. [2]Center for Emergent Matter Science, RIKEN, Wako, Saitama, Japan. [3]National Institute of Advanced Industrial Science and Technology (AIST), Research Institute for Hybrid Functional Integration, Tsukuba, Ibaraki, Japan. [4]Department of Physics, Pohang University of Science and Technology, Pohang, Korea. [5]Asia Pacific Center for Theoretical Physics, Pohang, Korea. [6]Department of Physics, Korea Advanced Institute of Science and Technology, Daejeon, Korea. [7]Institute of Physics, Johannes Gutenberg University Mainz, Mainz, Germany. [8]Peter Grünberg Institut, Forschungszentrum Jülich, Jülich, Germany. [9]Trans-scale Quantum Science Institute, The University of Tokyo, Bunkyo-ku, Tokyo, Japan. [10]These authors contributed equally: Weiguang Gao, Liyang Liao. ✉e-mail: yotani@issp.u-tokyo.ac.jp

overlap, potentially involving processes unrelated to OAM. This overlap could complicate the precise quantification of the interconversion between nonequilibrium OAM and charge current.

Nonlocal transport measurements[1,31–40] offer a promising approach to investigating various quantum transport phenomena, ranging from diffusive spin transport in metal[1,31–37] to unconventional spin conversion in moiré superlattices[41,42] as well as the quantum spin Hall[38] and quantum anomalous Hall effects[39,40]. In nonlocal transport devices, charge transport and nonequilibrium OAM can be spatially separated, enabling direct and isolated observation of the OAM distribution. Additionally, the nonlocal transport geometry naturally facilitates the measurement of both the direct (DOEE) and inverse (IOEE) orbital Edelstein effect within a single device, allowing verification of Onsager's reciprocal relation—a principle of fundamental importance in the study of quantum transport.

In this study, we present measurements of the lateral OAM distribution and the reciprocity of the orbital Edelstein effect using nonlocal transport devices comprising $Al_2O_3$, Cu, and ferromagnets (FMs). The nonequilibrium OAM can reach ~100 nm along a lateral direction. This decay length remains unaffected by the Cu thickness, suggesting the OAM distribution in the lateral direction differs from that in the vertical direction, which aligns with the formation of interfacial orbital Rashba states between Cu and oxides. The lateral decay length decreases with lower temperature, distinct from spin diffusion behavior. Crucially, all the experimental results comply with Onsager's reciprocal relations. Our work offers significant insights into the nonlocal correlation behaviors of nonequilibrium OAM and expands the design possibilities for orbitronics devices.

## Results and discussion

### Measurement of orbital accumulation in a nonlocal transport device structure

We investigated the OAM distribution using a nonlocal lateral transport structure[1,31–37] (Fig. 1a, b), where the nonlocal resistance signals arise from DOEE and IOEE. In our samples, an oxidized Cu layer (denoted by $CuO_x$ hereafter) <3 nm thick (Supplementary Section 1) forms between the $Al_2O_3$ and Cu layers due to natural oxidation (method). In the direct measurement configuration (Fig. 1a), a charge current $I_c$ is applied to the $Al_2O_3$/$CuO_x$/Cu nanowire oriented along the $y$ axis in Fig. 1a (denoted by $Cu_y$ nanowire hereafter). This current induces nonequilibrium OAM (L) polarized in the $x$-direction according to the OEE Hamiltonian, $H_{OEE} = \alpha_{OEE}(\mathbf{L} \times \mathbf{k}) \cdot \hat{\mathbf{z}}$, where $\alpha_{OEE}$ is the orbital Rashba coefficient, $\mathbf{k}$ is the wavevector, and $\hat{\mathbf{z}}$ is the unit vector along the $z$ axis. If a nonlocal orbital response occurs at the Cu/FM junction, it can be converted into spins through the SOC of the FM, generating an output voltage $V$ across the Cu/FM junction. We chose FM as the orbital converter instead of heavy metal so that we can separate the nonequilibrium OAM-induced signal from the bypass current-induced offset by varying the magnetization direction $\Phi$. The nonlocal signal as a function of the separation distance $d$ reflects the lateral distribution of the nonequilibrium OAM.

In the inverse measurement configuration, the processes are reversed according to Onsager's reciprocal relations (Fig. 1b). A charge current $I_c$ is injected into the $Al_2O_3$/$CuO_x$/Cu nanowire, which is oriented along the $x$ axis in Fig. 1b (denoted by $Cu_x$ nanowire hereafter) from the FM magnetized along the $x$ axis. This generates a nonequilibrium $x$-polarized OAM in the $Cu_x$ nanowire via the SOC of the FM,

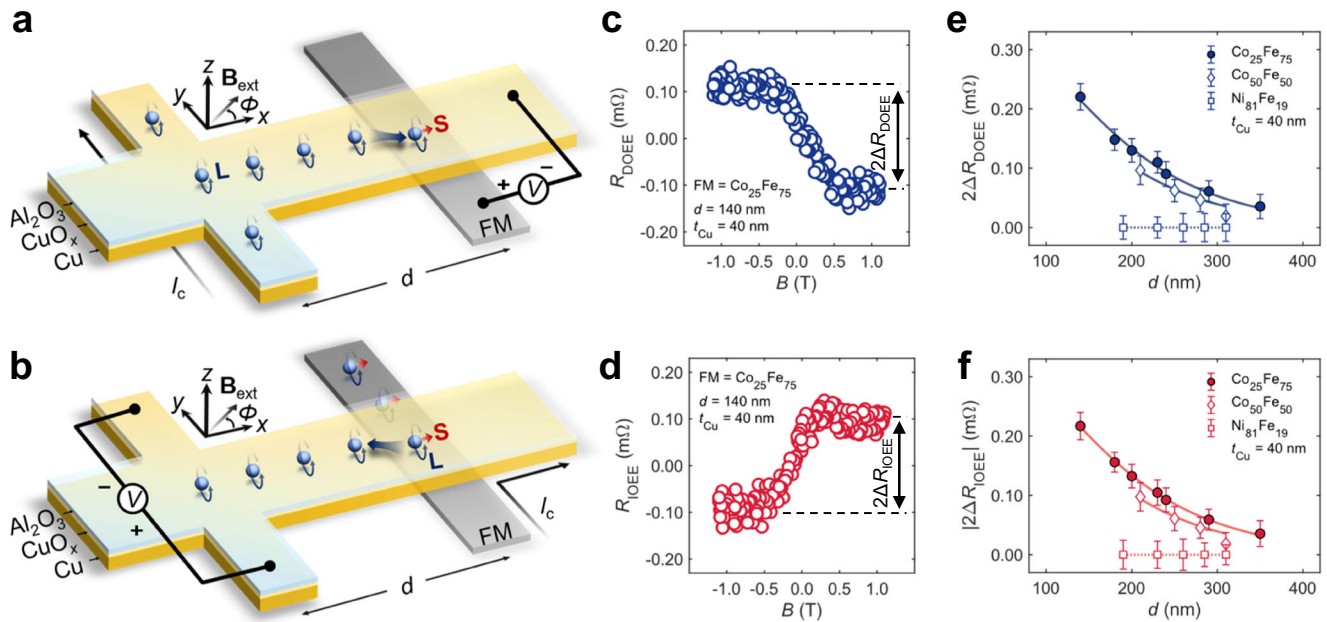

**Fig. 1 | Schematic illustration of nonlocal transport measurements and verification of orbital response through ferromagnetic materials dependence experiments. a** Nonlocal measurement configuration to observe DOEE (direct measurement). The nonequilibrium OAM (L) is generated by the charge current $I_c$ at the $CuO_x$/Cu interface through DOEE. The orbital accumulation then converts into SAM (S) via SOC in FMs and induces a nonlocal response $V$. **b** Nonlocal measurement configuration to observe IOEE (inverse measurement). A charge current brings nonequilibrium OAM from the FMs. The orbital accumulation then converts to charge current at the $CuO_x$/Cu interface through IOEE. In nonlocal measurement (**a, b**), the orbital generator and detector are sufficiently isolated in space with a separation distance $d$ (center-to-center distance), allowing the measurement of nonlocal orbital response. **c, d** Typical results of direct (**c**, $R_{DOEE}$) and inverse (**d**, $R_{IOEE}$) nonlocal orbital Edelstein resistance $R$, which is defined as $R \equiv V/I_c$. The

results are observed in sample A with separation distance $d = 140$ nm, Cu thickness $t_{Cu} = 40$ nm and FM = $Co_{25}Fe_{75}$ (Methods) while the external magnetic field $\mathbf{B}_{ext}$ swept along the hard axis of FM ($\Phi = 0°$) from −1.25 T to 1.25 T. All the signals are globally offset to position their center at $R = 0$ Ω. The double-headed arrows in **c, d** indicate the definition of $2\Delta R_{DOEE}$ and $2\Delta R_{IOEE}$, where $2\Delta R_{DOEE} = -2\Delta R_{IOEE} \approx 0.22$ mΩ. **e, f** FM dependence results of $2\Delta R_{DOEE}$ (**e**) and $2\Delta R_{IOEE}$ (**f**). The solid curves represent the fitting of the data to Eq. (1), implying a long-range decay length of orbital accumulation $\lambda_o$ of ~100 nm regardless of the selection of FMs. The dotted curves are guiding lines showing the value $R = 0$ Ω. The error bars indicate the standard deviation of $R$ after the magnetization is saturated. All the results are measured at room temperature. All the results are in solid agreement with Onsager's reciprocal relations.

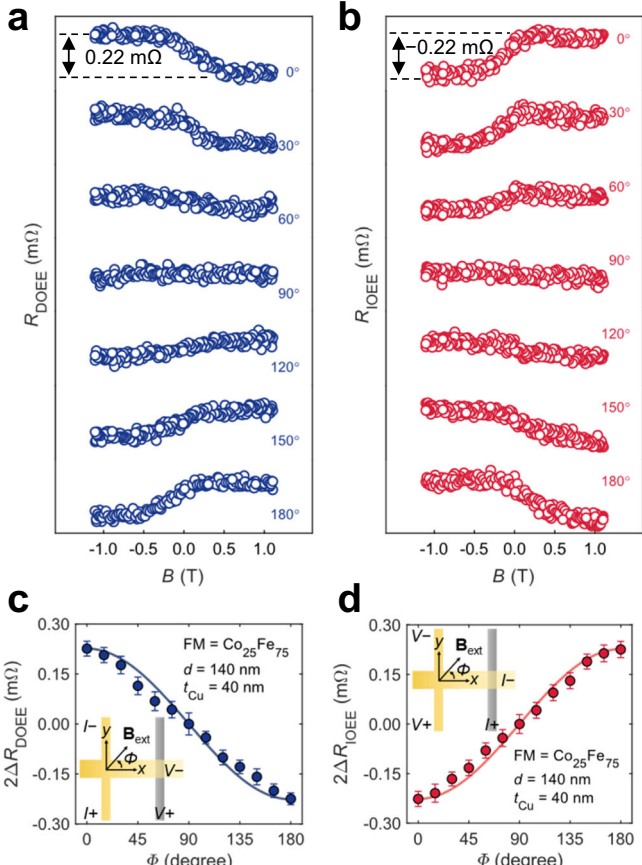

**Fig. 2 | Angle dependence experiments. a, b** $R_{DOEE}$ (**a**) and $R_{IOEE}$ (**b**) for sample A observed with magnetic field $\mathbf{B}_{ext}$ at various angle $\Phi$. **c, d** $2\Delta R_{DOEE}$ and $2\Delta R_{IOEE}$ as a function of $\Phi$. The insets show the measurement configuration and the definition of angle $\Phi$. The results are measured at room temperature. The solid curves in **c, d** are guiding lines employing a cosine function $f(\Phi) = f_1 \cos \Phi$. The angle-dependent $2\Delta R_{DOEE}$ and $2\Delta R_{IOEE}$ well match the cosine function, indicating that $x$-polarized OAM is measured. The error bars in **c, d** indicate the standard deviation of $R$ after the magnetization is saturated. All the results ascertain Onsager's reciprocal relations.

which is converted to a charge current through IOEE, resulting in the voltage signal $V_{IOEE}$. Spin current is also generated in this process; however, the negligible SOC in Cu[33] should only lead to a tiny contribution to the voltage signal, as will be demonstrated below.

We first demonstrate both direct and inverse measurements to validate the feasibility of measuring nonequilibrium OAM using a nonlocal configuration. Fig. 1c, d illustrate the typical direct ($R_{DOEE}$) and inverse ($R_{IOEE}$) orbital Edelstein resistance as a function of the magnetic field along the $x$ axis, observed in sample A (separation distance $d = 140$ nm, Cu thickness $t_{Cu} = 40$ nm and FM = Co$_{25}$Fe$_{75}$). The nonlocal orbital Edelstein resistance is defined as $R \equiv V/I_c$, where $R$ refers to both $R_{DOEE}$ and $R_{IOEE}$, $V$ is the measured voltage signal, and $I_c$ is the applied current. Both $R_{DOEE}$ and $R_{IOEE}$ exhibit a linear response to the external magnetic field $\mathbf{B}_{ext}$ within the range of −0.5 T to +0.5 T, saturating once the magnitude of $\mathbf{B}_{ext}$ exceeds the saturation field $\mathbf{B}_{sat}$ (given by the anisotropic magnetoresistance curve, see Supplementary Section 2) of magnetization. Taking the difference of $R_{DOEE}$ and $R_{IOEE}$ at +$\mathbf{B}_{ext}$ and −$\mathbf{B}_{ext}$, we obtained a $2\Delta R_{DOEE} = 0.22$ mΩ and $2\Delta R_{IOEE} = -0.22$ mΩ. The absolute value of the signal is two orders of magnitude larger than the artifact coming from the stray field-induced Hall effect in Cu, estimated via the COMSOL simulation (Supplementary Section 3).

As a typical way to provide evidence for nonequilibrium OAM[18,20,21,43,44], we conducted FM dependence experiments using Co$_{25}$Fe$_{75}$, Co$_{50}$Fe$_{50}$, and Ni$_{81}$Fe$_{19}$, as shown in Fig. 1e, f, where $2\Delta R$ is the overall change in nonlocal resistance. $|2\Delta R(Co_{25}Fe_{75})|$ is larger than $|2\Delta R(Co_{50}Fe_{50})|$, and $|2\Delta R(Ni_{81}Fe_{19})|$ is at least one order of magnitude smaller than them. A different measurement focusing on the local responses[45] also confirms the FM dependence (measurement configurations and results are detailed in Supplementary Section 4), where the orbital response is measured by FM nanowires right below the orbital generation part. Consistent with previous studies[18,20,21,43,44], the FM-dependent results identify the characteristic of orbital responses. We attribute the strong FM dependence to the spin-orbit correlation $\langle \mathbf{L} \cdot \mathbf{S} \rangle^{FM}$ within FMs[43,44,46]. $\langle \mathbf{L} \cdot \mathbf{S} \rangle^{FM}$ near the Fermi level in FMs varies significantly with slight changes in FM composition, leading to $\langle \mathbf{L} \cdot \mathbf{S} \rangle^{Co_{25}Fe_{75}} > \langle \mathbf{L} \cdot \mathbf{S} \rangle^{Co_{50}Fe_{50}} > \langle \mathbf{L} \cdot \mathbf{S} \rangle^{Ni_{81}Fe_{19}}$ [43]. As the stray fields of the Co$_{25}$Fe$_{75}$ and the Ni$_{81}$Fe$_{19}$ are in the same order, the large difference between $|2\Delta R(Co_{25}Fe_{75})|$ and $|2\Delta R(Ni_{81}Fe_{19})|$ also excludes the stray field-induced artifacts in the nonlocal resistance. The spin-induced nonlocal resistance also cannot strongly depend on the FM type (see Supplementary Section 5, where Co$_{25}$Fe$_{75}$ and Ni$_{81}$Fe$_{19}$ show comparable nonlocal spin valve signals), suggesting that the observed $\Delta R$ is dominated by OAM.

Next, we analyzed the nonlocal resistance as a function of the separation distance $d$. The nonlocal resistance decreases exponentially with increasing $d$ (Fig. 1f), indicating that the OAM accumulation decays as the detection point moves farther from the charge current source. A significant signal persists up to $d \sim 350$ nm, corresponding to an exponential decay length of approximately 100 nm. At the same time, a trivial current bypass effect can generate a nonlocal response, as previously reported[47,48]; both COMSOL simulations and analytical modeling estimate a decay length of approximately 47 nm (Supplementary Section 6). This is substantially shorter than the experimentally observed value of ~100 nm, thereby excluding the current bypass effect as the sole origin of the observed nonlocal response. Accounting for both OAM decay and bypass effects, we applied the following equation to evaluate the lateral decay length of orbital accumulation distribution (Supplementary Section 6):

$$2\Delta R = \frac{A}{\frac{1}{\lambda_o} - \frac{\pi}{W_x}} \left[ \exp\left( -\frac{\pi d}{W_x} \right) - \exp\left( -\frac{d}{\lambda_o} \right) \right], \quad (1)$$

where $A$ is the fitting parameter for local processes, including the vertical OAM decay, charge-to-orbital and orbital-to-spin conversion, $\lambda_o$ is the lateral decay length of orbital accumulation, and $W_x$ is the width of the Cu nanowire. As shown in Fig. 1e, f, fitting the data for Co$_{25}$Fe$_{75}$ (circles) and Co$_{50}$Fe$_{50}$ (diamonds) to Eq. (1) yields $\lambda_o \sim 100$ nm, regardless of the FM selection, suggesting that the nonlocal correlation is determined by the processes in the Al$_2$O$_3$/CuO$_x$/Cu nanowire and unaffected by the Cu/FM junctions.

Measuring nonequilibrium OAM using the nonlocal lateral transport structure demonstrates the feasibility of detecting OAM through the chemical potential, allowing the measurement of pure orbital accumulation. Onsager's reciprocal relations are then validated in the experiments: the relationship $2\Delta R_{DOEE} = -2\Delta R_{IOEE}$ holds across all the measurements. This highlights that the OAM degree of freedom is an active and essential factor in electronic transport, a consideration that has been overlooked in previous research.

## Angular dependence

We then performed an angular dependence experiment on sample A to clarify the orientation of OAM. As the angle $\Phi$ of $\mathbf{B}_{ext}$ to the $x$ axis varied in-plane from 0° to 180°, $R_{DOEE}$ (Fig. 2a) and $R_{IOEE}$ (Fig. 2b) evolved accordingly. Notably, $2\Delta R_{DOEE}$ (Fig. 2c) and $2\Delta R_{IOEE}$ (Fig. 2d) show a strong correlation with the corresponding cosine curves $f(\Phi) = f_1 \cos \Phi$, where $f_1$ represents the value of $2\Delta R_{DOEE}$ or $2\Delta R_{IOEE}$ at

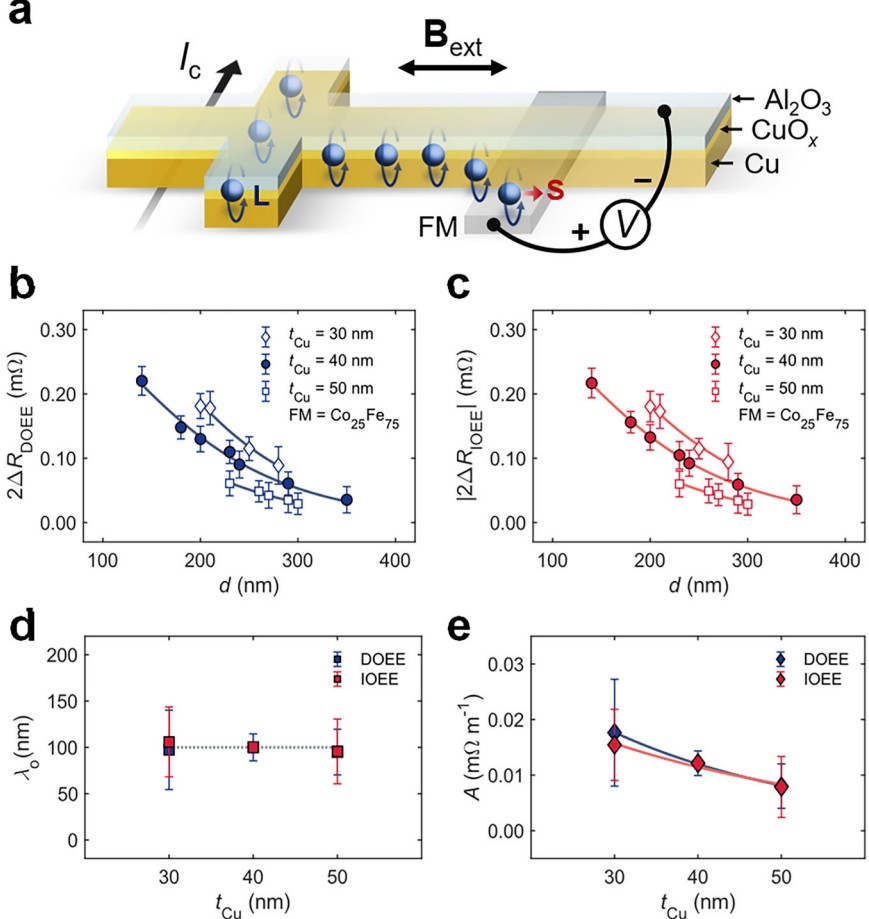

**Fig. 3 | Cu thickness dependence experiments. a** Schematic illustration of lateral and vertical OAM distribution and the role of oxidized and unoxidized Cu. The top Cu layer is oxidized and homogenous (CuO$_x$ region), assisting the long-range orbital response; the bottom Cu layer remains unoxidized (Cu region), exhibiting a short vertical decay length of orbital accumulation. In a nonlocal configuration, orbital accumulation distributes laterally and vertically. **b, c** Cu thickness dependence results of $2\Delta R_{DOEE}$ (**b**) and $2\Delta R_{IOEE}$ (**c**). The results are measured at room temperature with FM = Co$_{25}$Fe$_{75}$. The solid curves represent the fitting of the data to Eq. (1). The error bars in **b, c** indicate the standard deviation of $R$ after the magnetization is saturated. **d** $\lambda_o$ at various Cu thicknesses estimated from fitting, whose values are almost constant. The dotted line marks the value of $\lambda_o = 100$ nm. **e** Fitting result of parameter $A$. The solid curves show the exponential fitting for $A$. The vertical decay length of orbital accumulation ($\lambda_o^z \sim 25$ nm) is much smaller than the lateral one ($\lambda_o \sim 100$ nm), suggesting a distinction between lateral and vertical distribution of OAM. Error bars in **d, e** represent 95% confidence intervals from fitting results. All the results agree with Onsager's reciprocal relations.

90°. Analogous to the angle dependence in spin measurements[31], this strongly suggests that we are detecting OAM projected along the magnetization direction. This trend further implies that the measured OAM is polarized along the x-direction, consistent with OEE, perpendicular to the applied charge current[12,49]. Minor deviations from the cosine curve, where the data points vertically shift towards $2\Delta R = 0$ Ω, can be attributed to the magnetic anisotropy of the FM nanowire (Supplementary Section 7).

## Cu thickness dependence

To reveal both the lateral and vertical distributions of the OAM in the Al$_2$O$_3$/CuO$_x$/Cu nanowire system (Fig. 3a), we investigated the Cu thickness ($t_{Cu}$) dependence of the nonlocal signals, as summarized in Fig. 3b, c. Fitting $2\Delta R_{DOEE}$ (Fig. 3b) and $|2\Delta R_{IOEE}|$ (Fig. 3c) to Eq. (1), we observed that increasing $t_{Cu}$ causes only a slight change in $\lambda_o$ (Fig. 3d) but a significant reduction in $A$ (Fig. 3e). Considering that the thickness ($t_{CuO}$) of the oxidized Cu regions (CuO$_x$) remains the same in all the samples due to the identical atmospheric exposure times, almost no dependence of $\lambda_o$ on the Cu thickness suggests that oxidized Cu supports the lateral nonlocal orbital response. Conversely, the thickness of unoxidized Cu depends on $t_{Cu}$. We thus assumed $A$ contains the information on the vertical orbital distribution as $A = A'_{Cu} \exp(-t_{Cu}/\lambda_o^z)$

where $A'_{Cu}$ is a thickness-independent fitting parameter, and $\lambda_o^z$ is the vertical decay length of OAM. The fitting (Fig. 3e) yields $\lambda_o^z \approx 25$ nm. The Cu thickness dependence in the local configuration shows consistent results (Supplementary Section 8).

The distinction between $\lambda_o$ and $\lambda_o^z$ suggests different physical mechanisms governing the lateral and vertical OAM distribution in the oxide/Cu systems[21,22]. Previous studies on the vertical distribution of OAM[21,22,24] have highlighted the crucial role of oxidation. In oxidized Cu, the $3d$ electron shell is not fully occupied; orbital states allow electrons to host OAM. Such OAM can barely penetrate the unoxidized Cu region where the $3d$ electron shell is nearly fully occupied[50], leading to a short decay length. Meanwhile, the continuously oxidized Cu region with a specific thickness $t_{CuO}$ provides a uniform electronic band structure across the x-y plane, supporting orbital accumulation associated with the large $\lambda_o$.

## Temperature dependence

We next performed temperature-dependent experiments to gain deeper insight into the nonlocal orbital response. Fig. 4a shows $R_{DOEE}$ at 300 K (black squares) and 50 K (green circles) in sample A. The $R_{DOEE}$ signal is sizable at 300 K but vanishes at 50 K, demonstrating that the nonlocal orbital response diminishes at low temperatures. To

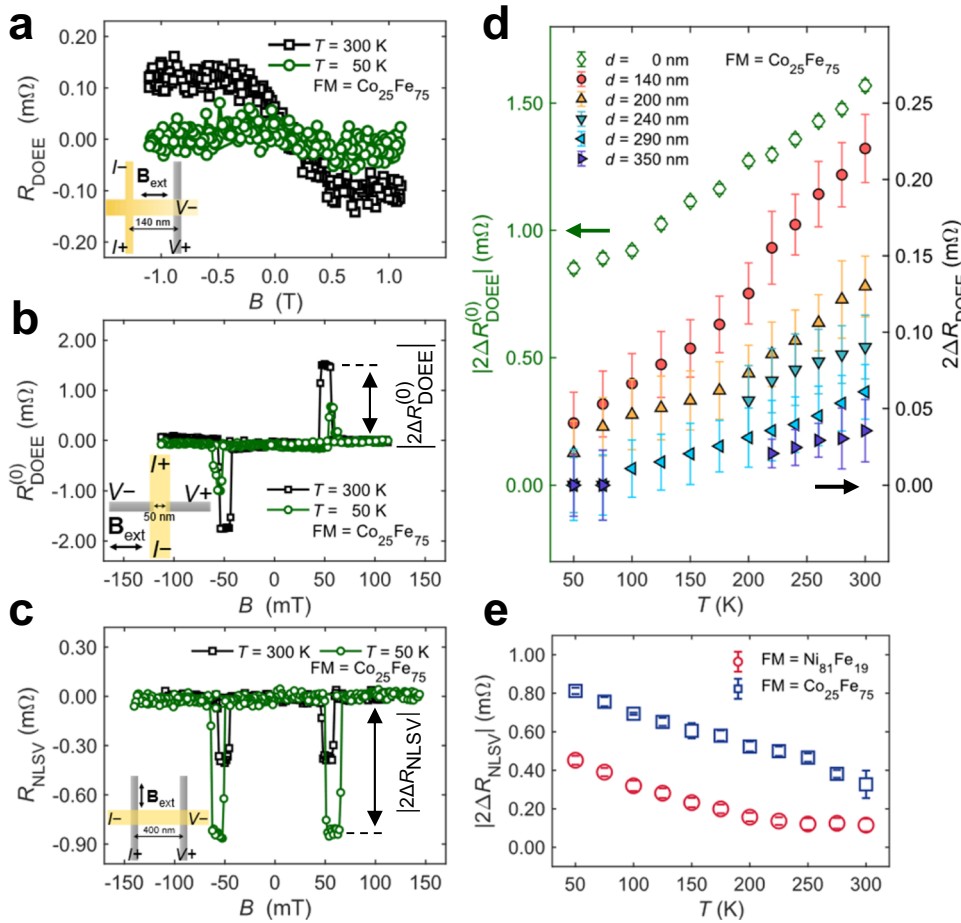

**Fig. 4 | Temperature dependence experiments. a–c** Typical signals for $R_{DOEE}$ (**a**), $R_{DOEE}^{(0)}$ (**b**), and $R_{NLSV}$ (**c**) at 300 K (black squares) and 50 K (green circles). $R_{DOEE}$ and $R_{DOEE}^{(0)}$ signals decrease with lower temperatures, while $R_{NLSV}$ signals increase as the temperature decreases. The double-headed arrows in **b**, **c** indicate the definition of $2\Delta R_{DOEE}^{(0)}$ and $2\Delta R_{NLSV}$. The insets in **a–c** represent the measurement configuration, respectively. **a–c** $t_{Cu} = 40$ nm and FM = $Co_{25}Fe_{75}$. **d** Temperature dependence results of $2\Delta R_{DOEE}^{(0)}$ (open diamond markers with green color, left y axis) and $2\Delta R_{DOEE}$ (circle and triangle markers filled with other colors, right y axis). The error bars indicate the standard deviation of $R$ in the plateau. The results are measured in

samples with $t_{Cu} = 40$ nm and FM = $Co_{25}Fe_{75}$. All the $2\Delta R_{DOEE}^{(0)}$ and $2\Delta R_{DOEE}$ decrease with lowering temperature. **e** Summarized temperature dependence results of $2\Delta R_{NLSV}$ with $Co_{25}Fe_{75}$ (open blue squares) and $Ni_{81}Fe_{19}$ (open red circles). The error bars indicate the standard deviation of $R_{NLSV}$ in the plateau. $2\Delta R_{NLSV}$ increases with lowering temperature. The spacing distance between two FM nanowires is $d_{NLSV} = 400$ nm for both $Ni_{81}Fe_{19}$ and $Co_{25}Fe_{75}$ samples (Methods). The opposite temperature dependence on the behaviors of DOEE (**d**) and NLSV (**e**) implies distinct physics between orbital accumulation and spin accumulation.

disentangle the local and nonlocal contributions during the temperature evolution, we also performed temperature-dependent measurements in the local configuration[45], where the orbital response is probed by FM nanowires located below the orbital generation region, as shown in Fig. 4b. At 300 K (black squares), the local $R_{DOEE}^{(0)}$ signal is also larger than that at 50 K (green circles), although half of the signal remains at 50 K. These behaviors are in sharp contrast with the spin transport in a nonlocal spin valve (NLSV) structure[31–34], characterized by $R_{NLSV}$ (Fig. 4c), where the signal at 50 K is larger (green circles) than that at 300 K (black squares).

Fig. 4d summarizes the temperature dependence of the local $2\Delta R_{DOEE}^{(0)}$ (green circles) and the nonlocal $2\Delta R_{DOEE}$ at different distances $d$ (other different colors). The corresponding raw data are presented in Supplementary Section 9 (Figs. S12 and S13). The signal decreases at low temperatures in both local and nonlocal measurements. At 50 K, while the nonlocal signals drop to nearly zero, the local signals still retain a finite value, indicating that the local signal decreases more slowly than the nonlocal signal. The same temperature dependence is observed in the inverse measurement (Fig. S14), and no signal is observed at any temperature in the nonlocal orbital device with FM = $Ni_{81}Fe_{19}$ (Fig. S15). Meanwhile, the nonlocal spin signal $2\Delta R_{NLSV}$

increases with decreasing temperature (Fig.4e) and is present in both devices with FM = $Co_{25}Fe_{75}$ and $Ni_{81}Fe_{19}$, clearly indicating the different physics involved in the observed nonlocal orbital and spin signals (also see Supplementary Section 10).

To isolate the contribution of the nonlocal process in the temperature dependence of $2\Delta R_{DOEE}$, we assume that the local process parameter $A$ for the nonlocal signal $2\Delta R_{DOEE}$ in Eq. (1) has the same temperature dependence as the local signal $2\Delta R_{DOEE}^{(0)}$. This leads to the relationship $A(T)/A(300K) = \Delta R_{DOEE}^{(0)}(T)/\Delta R_{DOEE}^{(0)}(300K)$, allowing us to calculate the $A(T)$ values at different temperatures using $A(300K)$ (Fig. 1e, f) and the temperature dependence of the local signal. We then fitted the distance $d$ dependence of $2\Delta R_{DOEE}$ (Fig. 5a) and $|2\Delta R_{IOEE}|$ (Fig. S16) at each temperature using Eq. (1) and $A(T)$ given above so that $\lambda_o$ becomes the only fitting parameter as summarized in Fig. 5b. Interestingly, $\lambda_o$ decreases as temperature drops, which contrasts with the typical behavior of spin transport in metals, where diffusion length is longer at lower temperatures[32]. The temperature-dependent $\lambda_o$ also contrasts with the bypass effect, which remains unchanged at different temperatures[47,48] (Supplementary Section 11). This suggests that different physical mechanisms mediate the nonlocality of OAM. Calculations have shown that the orbital diffusion length in

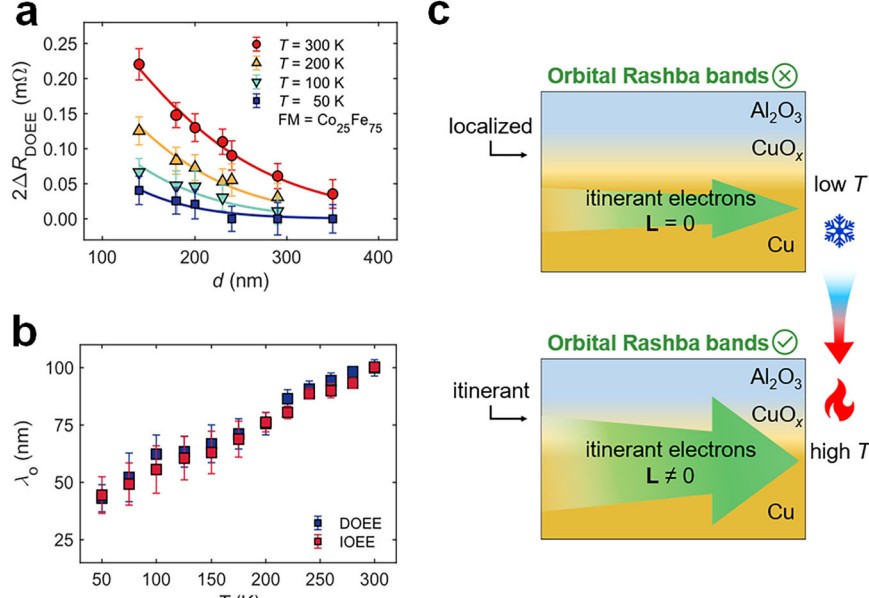

**Fig. 5 | Analysis on temperature-dependent orbital transport. a** Fitting of the temperature-dependent $2\Delta R_{DOEE}$ data with Eq. (1). Here, $t_{Cu}=40$ nm and FM = Co$_{25}$Fe$_{75}$. The solid curves represent the fitting results. The error bars indicate the standard deviation of $R$ after the magnetization is saturated. **b** Fitting result of $\lambda_o$ for $2\Delta R_{DOEE}$ (blue squares) and $2\Delta R_{IOEE}$ (red squares) as functions of temperature. The error bars indicate the 95% confidence intervals from fitting results. All the results obey Onsager's reciprocal relations. **c** Schematic of the proposed mechanism for the temperature-dependent orbital propagation. At low temperatures (upper panel), hopping between the localized states in the oxidized Cu (CuO$_x$ layer) is limited, so the itinerant electrons (green arrow) are confined to the unoxidized Cu layer without carrying OAM ($\mathbf{L}=0$). Thus, continuous orbital Rashba bands are not forming. At high temperature (lower panel), the localized states are thermally broadened, establishing the conductive channel in the CuO$_x$ layer. The electron wavefunctions are extended into the CuO$_x$ layer, creating hybridized states with OAM ($\mathbf{L}\neq0$), which form the continuous orbital Rashba bands and allow the long-range orbital propagation.

centrosymmetric systems is often below 10 nm[51]. Therefore, the non-locality may not be linked to a diffusive orbital current. Still, rather due to the eigenstates in the Rashba band, which host OAM by themselves, the Rashba effect locks the nonequilibrium OAM locally to the diffusive linear momentum[52,53] of the electrons. We propose a possible picture for the temperature dependence as shown in Fig. 5c. At low temperatures, the limited hopping between the localized states in the oxidized Cu cannot support orbital Rashba bands. The itinerant electrons have wavefunctions constrained within the unoxidized Cu layer without carrying OAM. The OAM propagation is more likely to occur in the high-temperature region, where free carriers can be excited due to the thermal broadening of localized states, and a conductive channel is formed at the oxidized Cu layer (Supplementary Section 12). As a result, the itinerant electrons have wavefunctions extending into the oxidized Cu layer, creating hybridized states with OAM. Thus, continuous orbital Rashba bands are formed on the whole Al$_2$O$_3$/CuO$_x$/Cu nanowire, and long-range orbital propagation is allowed.

In summary, our study investigates the lateral distribution of nonequilibrium OAM and the reversible OEE by utilizing nonlocal response in an orbital Edelstein system composed of Al$_2$O$_3$, Cu, and ferromagnetic materials. The nonlocal response is observed ~100 nm away from the injected electric current at room temperature, with its orbital nature confirmed by FM dependence and the polarization direction verified through angular dependence. The decay length $\lambda_o$ of orbital accumulation is unaffected by Cu thickness, suggesting that the oxidized Cu region assists the long-range nonlocal orbital response. In contrast to the spin diffusion length, $\lambda_o$ decreases with decreasing temperature, indicating that mechanisms other than diffusive orbital current mediate OAM transport. Importantly, in all the experiments, the orbital and charge accumulation induced by DOEE and IOEE are observed in the same devices and are mathematically equivalent, consistent with Onsager's reciprocal relations. Our work provides clear insights into the intrinsic reciprocal relationship of orbital effects. It reveals a long-range lateral correlation of OAM accumulation, paving the way for inter-connectable orbitronics devices capable of operating over long distances.

## Methods

### Sample fabrication

All devices were microfabricated on SiO$_2$/Si substrates using electron-beam lithography with polymethyl-methacrylate resist, followed by development, deposition, and lift-off processes. The FM nanowires were electron-beam evaporated on SiO$_2$/Si substrates. The Cu nanowires were deposited across the FM nanowires using a Joule heating evaporator. Before Cu deposition, the FM surface was carefully Ar-ion milled to obtain a transparent interface between the FM and Cu. The samples were exposed to the atmosphere at room temperature for 10 minutes prior to Al$_2$O$_3$ deposition to achieve a naturally oxidized Cu layer. The Al$_2$O$_3$ capping layers to prevent Cu from further oxidation were only deposited on the Cu nanowires by electron-beam deposition. All fabrication processes were performed at room temperature, with a base pressure of deposition lower than $5.0\times10^{-8}$ Torr and a pressure during deposition lower than $1.0\times10^{-7}$ Torr.

Unless otherwise specified, all the samples are fabricated in an identical and carefully designed geometry. In all the samples, the FM nanowires were composed of Co$_{25}$Fe$_{75}$, Co$_{50}$Fe$_{50}$, or Ni$_{81}$Fe$_{19}$, with a thickness of 20 nm ($t_{FM}$) and a width of 100 nm ($w_{FM}$). All Cu nanowires are 40 nm thick ($t_{Cu}$), with additional values of 30 nm and 50 nm used in the Cu thickness dependence experiment. The thickness of the Al$_2$O$_3$ capping layer is 15 nm for all the samples.

In nonlocal orbital transport devices (shown in Figs. 1a, b, and 4a), the cross-shaped Cu nanowires were fabricated where the Cu nanowires lying on the $y$ axis (Cu$_y$, parallel to FM nanowire) are 100 nm wide. The one lying on the $x$ axis (Cu$_x$, perpendicular to FM nanowire) is 150 nm wide. The center-to-center separation distance, $d$, between Cu$_y$ and FM, varies from 140 to 350 nm.

In local orbital transport devices (Fig. 4b), the FM nanowires are aligned collinearly with a 50 nm spacing gap. The 200 nm wide straight Cu nanowires are deposited perpendicular to the FM pair to bridge the gap.

In nonlocal spin valve devices (Fig. 4c), two parallel FM nanowires are separated by a fixed center-to-center spacing $d_{NLSV} = 400$ nm (for both $Ni_{81}Fe_{19}$ and $Co_{25}Fe_{75}$ samples). The Cu nanowires bridging the FM pairs are 150 nm in width.

## Electrical measurements

All the electric transport measurements were carried out in a He flow cryostat using the lock-in measurement technique. Electrical contacts to devices were made with wire bonding with Al wires. The applied alternating currents were calibrated to 500 μA for all measurements. An in-plane external magnetic field, $\mathbf{B}_{ext}$, was applied during all measurements. In nonlocal orbital transport measurements, $\mathbf{B}_{ext}$ was aligned along the hard axis of the FM nanowire. In angle-dependent measurements, the $\mathbf{B}_{ext}$ was rotated from the $x$ axis by the angle $\Phi$ (0°–180°). In nonlocal spin valve and local orbital transport measurements, $\mathbf{B}_{ext}$ was applied along the easy axis of the FM nanowires.

## Data availability

All the data supporting the findings of this study are available within the article and its Supplementary Information. The data generated in this study are provided in the Source data are provided with this paper.

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

## Acknowledgements

We thank Thierry Valet for the fruitful discussions and helpful comments. We also thank Shoya Sakamoto, Jieyi Chen and Ullerithodi Manaal Resivi for assistance. L.L. would like to thank the support from JSPS through "Research program for Young Scientists" (No. 23KJ0778). J.K. and Y.O. appreciate the financial support from Japan Society for the Promotion of Science (JSPS) KAKENHI (Grants No. JP19K05258, No.JP23K04574, and No. JP19H05629). Y.O. appreciates the financial support from Japan Science and Technology Agency (JST), Adopting Sustainable Partnerships for Innovative Research Ecosystem (ASPIRE), No. JPMJAP2410. H.-W.L. was supported by the Samsung Science and Technology Foundation (BA-1501-51) and the National Research Foundation of Korea (NRF) grant funded by the Korean government (MSIT) (No. RS-2024-00410027). K.-J.L. was supported by the National Research Foundation of Korea (NRF) grant funded by the Korean government (MSIT) (No. 2022M3H4A1A04096339). D.G. and Y.M. gratefully acknowledge financial support by the Deutsche Forschungsgemeinschaft (DFG, German Research Foundation)—TRR 288/2 – 422213477 (project B06), TRR 173/3–268565370 (project A11), and by the EIC Pathfinder OPEN grant 101129641 "OBELIX".

## Author contributions

Y.O., H.I., W.G., and J.K. conceived the experiment. W.G. performed the experiment with the help of H.I. and S.M. L.L., W.G., J.K., and Y.O. analyzed the experimental data. W.G. and N.B. performed COMSOL simulations. L.L. developed the theoretical models with the help of H.-W.L. D.G., Y.M., and K.-J.L. asserted the models. Y.O. supervised the project. L.L., W.G., and Y.O. wrote the manuscript with input from all the authors. All the authors reviewed and revised the manuscript. W.G. and L.L. contributed equally to the whole work.

## Competing interests

The authors declare no competing interests.
