## [Transparent Peer Review file · Nature Communications]

Nonlocal Electrical Detection of Reciprocal Orbital Edelstein Effect

Corresponding Author: Professor Yoshichika Otani

Version 0:

Reviewer comments:

Reviewer #1

(Remarks to the Author)

The authors report experiments dedicated to elucidating the direct and inverse orbital Edelstein effect by means of non-local measurements in the Al₂O₃/Cu system. Orbital angular momentum is generated by running a current through a narrow Al₂O₃/Cu wire. The non-local orbital response is then converted into spin through the spin-orbit coupling in the FM contact. The experiments demonstrate that when the sample is subjected to in-plane magnetic fields along the hard axis of the FM electrode, the non-local response increases linearly, saturating when the magnetization of the FM electrodes aligns along its in-plane hard axis.

While I believe the experimental data are of good quality and deserve publication, I have doubts about the interpretation of the data, mostly due to the geometry of the device together with the use of a FM contact to observe the effect. Therefore, I am not fully convinced yet about the authors' interpretation, which leads me to doubt about the significance of this work as well as whether this manuscript satisfies the criteria for publication in Nature Communications. I would be in favor of accepting publication if the authors can address all my concerns, which I detail below.

** It is not sufficiently clear why, if the non-local orbital response is converted into spin through spin-orbit coupling (SOC), the authors use an FM contact as a detector (or injector in the case of the reciprocal effect) instead of a spin-orbit heavy metal. Considering the short channel length of the devices, when the FM rotates its magnetization towards its hard axis, it creates stray fields at the injector (detector) region, which naturally produce a Hall signal in the Cu nanowire detection region. The use of a heavy metal should also work and will avoid the presence of such stray fields, which result in artifacts. I am pretty sure about the presence of such stray fields that the authors are not considering either in their experiments or in their simulations.

** Authors assume that the spin current generated when the current is injected through the FM plays a minimal role. I do not see any proof of that in their measurements, nor is the generated spin current negligible in all the temperature range measured. This could be easily verified through a non-local spin valve geometry by adding a second FM contact to the right side of the first FM. This should not be a problem for the authors, who are world-leading in the electrical characterization of such metallic spin valve devices.

** Regarding Cu thickness-dependent experiments: Figure 3 shows an oxidized Cu layer between Al₂O₃ and non-oxidized Cu. According to the sample fabrication protocol described in the methods section, this oxidized Cu layer should be present in all the fabricated devices. However, when looking at Figure 1, this oxidized layer of Cu is not represented, and the authors state that the orbital angular momentum is created at the Al₂O₃-Cu interface (caption of Figure 1). This is somewhat confusing that must be clarified, particularly because the authors are considering an interfacial effect. I believe the orbital angular momentum is created at the oxidized-non-oxidized Cu interface and not at the Al₂O₃-Cu interface, which indeed does not exist in their samples. Could the authors be more precise in explaining this? Also, how thick is the oxidized Cu layer formed by natural oxidation in air during 10 minutes?

Reviewer #2

(Remarks to the Author)

The manuscript "Nonlocal Electrical Detection of Reciprocal Orbital Edelstein Effect" presents an experimental investigation of the Onsager reciprocity in orbital transport using nonlocal transport measurements. The authors use an $\text{Al}_2\text{O}_3/\text{Cu}$ /ferromagnet structure to test the reciprocal relation between direct and inverse orbital Edelstein effects (DOEE and IOEE). Their results demonstrate that reciprocal orbital responses produce equivalent electric voltages, supporting Onsager reciprocity. The authors also analyze the decay length of orbital accumulation with variations in Cu thickness and temperature, concluding that the behavior contrasts with spin transport dynamics.

This work addresses a largely unexplored aspect of orbitronics, specifically in nonlocal OAM transport, and the findings have the potential to advance the understanding of OAM dynamics. The experimental design is robust, and the analyses are well-detailed, though certain points could benefit from clarification and further examination.

Specific Comments:

1) Current Bypass Effect Explanation: The authors consider that the current bypass effect cannot account for the observed nonlocal response, as their simulations indicate a bypass effect decay length of 50 nm, compared to the experimental 100 nm. However, the difference between these two decay lengths may not be large enough to definitively rule out the bypass effect. It would strengthen the argument if the authors addressed the temperature dependence of the bypass effect to further separate it from the orbital accumulation. Such an analysis could clarify whether the observed temperature-dependent decay behavior is indeed related to the orbital effects or influenced by bypass effects.

2) Mechanism of Long-Range Transport in Oxidized Cu: A critical element in the authors' interpretation is that the oxidized Cu layer enables long-range orbital transport due to its unique electronic band structure. This assumption is fundamental to explaining the observed 100 nm decay length, but supporting evidence is limited. Providing further theoretical or empirical support for this mechanism would be helpful, as it is essential to distinguish the orbital response from other possible contributions. In fact, it could be beneficial to consider if a temperature-dependent mechanism like variable range hopping (VRH) might also influence transport in the oxidized Cu layer. VRH, often associated with transport in disordered systems, involves charge carriers hopping between localized states, a process that increases with temperature. If the transport in the oxidized Cu is dominated by VRH, the current through this layer could depend on temperature and, consequently, affect the OREE as well.

Examining whether VRH could be active in the oxidized Cu layer would be particularly relevant for interpreting the temperature dependence observed in the orbital accumulation. While it may not directly explain the differing decay lengths, VRH could provide insights into how temperature-dependent transport in the oxide influences overall current flow and orbital effects. If feasible, an analysis or discussion on the potential influence of VRH on orbital accumulation (or other mechanisms that the authors find relevant) would help clarify the role of the oxide layer in the observed temperature dependence.

Reviewer #3

(Remarks to the Author)

In this study, the author experimentally tested the Onsager reciprocity of orbital transport in an orbital Edelstein system by employing non-local measurements. This research provides a clear understanding of the intrinsic reciprocity relationship of orbital effects and reveals the long-range transverse correlation of orbital angular momentum accumulation, laying a foundation for the development of interconnected orbital electronics devices that can be operated over long distances. It's so interesting. However, there are some issues that need to be addressed before formal publication.

1. Although the spin Hall effect emphasizes that the material requires a large spin-orbit coupling strength, the spin-orbit coupling effect of the material itself can not be ignored. Thus, how can we determine whether the measurement result is due to the contribution of the orbital Hall effect, the spin Hall effect, or both?
2. Why is there a strong dependence on ferromagnetic materials, suggesting that the measured signals originate from orbital responses rather than spin responses?
3. Some pictures are too blurry, and the colors are too similar to distinguish. For example, figures 1(a) and 1(b), and figures 4(c) and 4(d).

Version 1:

Reviewer comments:

Reviewer #1

(Remarks to the Author)

I have carefully read the authors' response to my concerns. While I am mostly convinced by their answers, there is still a critical issue regarding the CuOx layer that I am not fully persuaded by.

The formation of a 2–3 nm CuOx layer -- which is a crucial ingredient for observing the Orbital Hall Edelstein Effect (OHEE)-- is supported by STEM and EDX measurements. However, I honestly do not clearly see how the authors conclude that this layer is precisely 2–3 nm thick. With the current EDX measurements, the resolution does not seem sufficient to make such a claim. Additionally, there is a non-negligible concentration of Al in the region where Cu and oxygen may form the oxide, which could further complicate the interpretation.

I believe there is a very simple yet effective experiment the authors should perform using a reference device, i.e., pure Cu capped with Al, where no Cu oxidation occurs. In this case, the observed OHEE, attributed to the CuOx/Cu interface, should

disappear and therefore not be detected. If the authors can perform this experiment and demonstrate that the effect is absent, I would be fully convinced of the validity of their observations.

Reviewer #2

(Remarks to the Author)

Following the revisions, the authors have significantly improved the manuscript by incorporating additional spin transport measurements, clarifying the roles of stray fields and bypass currents with extra COMSOL numerical calculations, and constructing a simplified theoretical model to explain the temperature dependence of the orbital response. These additions address the key concerns raised in the initial round of reports, strengthening the validity of their conclusions.

I now find the manuscript suitable for publication in its current form.

Reviewer #3

(Remarks to the Author)

As I have mentioned in my first report, I consider the results that are presented in this manuscript very interesting and I believe that this result has the potential to grow into a new research direction in orbitronics and related materials. The concerns that I had for the first version of the manuscript have been answered in a satisfactory manner so I would recommend this manuscript to be published in Nature Communications.

Version 2:

Reviewer comments:

Reviewer #1

(Remarks to the Author)

N/A

Thank you very much for your recent email. We very much appreciate the positive evaluation of our manuscript (NCOMMS-24-63599-T) by the reviewers “the experimental data are of good quality and deserve publication”, “This work addresses a largely unexplored aspect of orbitronics, specifically in nonlocal OAM transport, and the findings have the potential to advance the understanding of OAM dynamics”, and “This research provides a clear understanding of the intrinsic reciprocity relationship of orbital effects and reveals the long-range transverse correlation of orbital angular momentum accumulation, laying a foundation for the development of interconnected orbital electronics devices that can be operated over long distances”. Their comments allowed us to increase further the quality of the manuscript; we are therefore most grateful for their feedback.

Please find in the text below the detailed reply to each individual comment of the reviewers, and the main modifications we have made to our main manuscript and supplementary information.

Yours sincerely,

Y. Otani (on behalf of the authors)

Main modifications:

1. We added COMSOL simulation to show that the stray field-induced Hall effect contribution is two orders smaller than the observed signal (Supplementary section 3), excluding this artifact as the main source for the observed signal.
2. We performed additional nonlocal spin valve (NLSV) measurements to show

that spin current exists, but does not converted into charge current at the Rashba interface (Supplementary section 5). We also showed that the temperature dependence of spin transport is different from that of orbital transport (modified Fig. 4).

3. We showed that the bypass current has nearly no temperature dependence (Supplementary section 10).

4. We did additional analysis of the temperature dependence of the orbital propagation (modified Fig. 5 and Supplementary section 11).

5. We modified the previously blurry figures to enhance color contrast and improve data point distinguishability (Fig. 1, Fig. 4 and Fig. 5).

6. We performed cross-sectional TEM observations to determine the oxygen distribution near the Cu/Al₂O₃ interface (Supplementary section 1) and modified the schematics about the sample structure in Fig. 1.

Response to Reviewer #1:

The authors report experiments dedicated to elucidating the direct and inverse orbital Edelstein effect by means of non-local measurements in the Al₂O₃/Cu system. Orbital angular momentum is generated by running a current through a narrow Al₂O₃/Cu wire. The non-local orbital response is then converted into spin through the spin-orbit coupling in the FM contact. The experiments demonstrate that when the sample is subjected to in-plane magnetic fields along the hard axis of the FM electrode, the non-local response increases linearly, saturating when the magnetization of the FM electrodes aligns along its in-plane hard axis.

While I believe the experimental data are of good quality and deserve publication, I have doubts about the interpretation of the data, mostly due to the geometry of the device together with the use of a FM contact to observe the effect. Therefore, I am not fully convinced yet about the authors' interpretation, which leads me to doubt about the significance of this work as well as whether this manuscript satisfies the criteria for publication in Nature Communications. I would be in favor of accepting publication if the authors can address all my concerns, which I detail below.

Response: We sincerely thank the reviewer for the thorough evaluation and insightful comments. We provide detailed responses to each suggestion and comment below.

** It is not sufficiently clear why, if the non-local orbital response is converted into spin through spin-orbit coupling (SOC), the authors use an FM contact as a detector (or injector in the case of the reciprocal effect) instead of a spin-orbit heavy metal. Considering the short channel length of the devices, when the FM rotates its magnetization towards its hard axis, it creates stray fields at the injector (detector) region, which naturally produce a Hall signal in the Cu nanowire detection region. The use of a heavy metal should also work and will avoid the presence of such stray fields, which result in artifacts. I am pretty sure about the presence of such stray fields that the authors are not considering either in their experiments or in their simulations.

Response: We used an FM contact instead of a spin-orbit heavy metal so that we can control the orbital-spin conversion through the external magnetic field. A spin-orbit heavy metal does convert orbital accumulation into spin and produces a voltage signal, but the signal is unchangeable once the device is fabricated. Hence, considering the offset signal due to the small bypass current,

the orbital accumulation-induced signal cannot be separated from the measurement result. However, an FM contact allows us to detect orbital accumulation-induced voltage signal with opposite sign under opposite magnetic field and separate the orbital accumulation signal from the bypass current-induced offset.

We added following discussion in Page 5, Line 91 in the main text: **“We chose FM as the orbital converter instead of heavy metal, so that by varying the magnetization direction Φ , we can separate the nonequilibrium OAM-induced signal from the bypass current-induced offset.”**

As the reviewer pointed out, the stray field may also induce a signal via the Hall effect in Cu. But the Hall effect requires a transverse current, locally at the point being considered, i.e., not the stray fields itself, but the coexistence of the stray fields and the bypass current in the Cu nanowire detection region can produce a Hall signal. Furthermore, only when the stray field component, the bypass current and the detection Cu nanowire are orthogonal to each other, the Hall effect can contribute to the signal.

We therefore perform COMSOL simulation for the stray field and bypass current distribution in our device and estimate the upper limit of the Hall effect contribution. We added Section 3 in the Supplementary Information, including the following key points:

(1). We focus on the leading order contribution from the Hall effect, whose existence relies on the homogeneous parts of the $\mathbf{B}_{\text{stray}}$ and \mathbf{j}_{by} within a cuboid of the Cu nanowire and ignore the higher order contribution that relying on the inhomogeneity of the $\mathbf{B}_{\text{stray}}$ and \mathbf{j}_{by} within a cuboid of the Cu nanowire.

(2). In the direct measurement, the detected signal V reflects the potential difference between FM terminal and the right arm of Cu_x nanowire (Fig. S4a), which include the vertical potential difference (V_z) at the Cu/FM junction and x -direction potential (V_x) difference between the Cu/FM junction and right end of Cu_y nanowire. Thus, only in the region B where $\mathbf{B}_{\text{stray}}$ and \mathbf{j}_{by} satisfies

$$\textcircled{a} \sim -\hat{z} \times \hat{x} \parallel \hat{y} \text{ direction}$$

(Fig. S4b) and in the region A where $\mathbf{B}_{\text{stray}}$ and \mathbf{j}_{by} satisfies

$$\textcircled{b} \sim \hat{z} \times \hat{x} \parallel \hat{y} \text{ direction}$$

(Fig. S4c), the Hall effect contributes to the detected signal V .

Fig. S4 | Hall voltage induced by stray field and bypass current in direct measurement. **a**, Schematic illustration of the stray field and bypass current in the direct measurement setup. The bypass current density (ϕ_{by}) is represented by blue dashed curves, while the stray field (\mathbf{B}_{stray}) is shown in orange curves. Specific points along the centerline of the Cu_ξ nanowire, labeled as A, B, C, D, and E, are highlighted for further analysis. **b**, \mathbf{B}_{stray} and ϕ_{by} at the region near B point (above Cu/FM junction). A negative ζ -direction Hall electric field is induced, which can be detected by voltmeter. **c**, \mathbf{B}_{stray} and ϕ_{by} at the region near A point (on the right of Cu/FM junction). An ξ -direction Hall electric field is induced, which can be detected by voltmeter. **d**, **e**, The analysis method for COMSOL simulation. The bulk of Cu above FM (**d**) and that to the right of FM (**e**) is divided into 4 volume elements each. For each volume element, \mathbf{B}_{stray} and ϕ_{by} are simulated and averaged in each volume while the Hall voltage is calculated. This method provides a more accurate estimation by summing the contributions from all volume elements, compared to calculations based solely on a single point.

(3). In the inverse measurement, the measured signal V reflects the y -direction potential difference (V_y) between two ends of Cu_y nanowires (Fig. S5a). Thus, only in the region B where $\mathbf{B}_{\text{stray}}$ and \mathbf{j}_c satisfies $\mathbf{j}_c \sim \frac{1}{\mu_0} \nabla \times \mathbf{B}_{\text{stray}}$ (Fig. S5b) and in the region A where $\mathbf{B}_{\text{stray}}$ and \mathbf{j}_c satisfies $\mathbf{j}_c \sim -\frac{1}{\mu_0} \nabla \times \mathbf{B}_{\text{stray}}$ (Fig. S5c), the Hall effect contributes to the detected signal V .

Fig. S5 | Hall voltage induced by stray field and bypass current in inverse measurement. **a**, Schematic illustration of the stray field and bypass current in the inverse measurement setup. The bypass current density (j_{by}) is represented by blue dashed curves, while the stray field ($\mathbf{B}_{\text{stray}}$) is shown in orange curves. Specific points along the centerline of the Cu_x nanowire, labeled as A, B, C, D, and E, are highlighted for further analysis. **b**, $\mathbf{B}_{\text{stray}}$ and \mathbf{j}_{by} at the region near B point (above Cu/FM junction). A y -direction Hall electric field is induced, which can be detected by voltmeter. **c**, $\mathbf{B}_{\text{stray}}$ and \mathbf{j}_{by} at the region near A point (on the right of Cu/FM junction). A y -direction Hall electric field is induced, which can

be detected by voltmeter. d, e, The analysis method for COMSOL simulation. The bulk of Cu above FM (d) and that to the right of FM (e) is divided into 4 volume elements each. For each volume element, B_{stray} and j_{by} are simulated and averaged in each volume while the Hall voltage is calculated. This method provides a more accurate estimation by summing the contributions from all volume elements, compared to calculations based solely on a single point.

Table 1

Direct	Ⓜ වෘත්ත	ආ වෘත්ත	ආ වෘත්ත
රිම/Ⓜ	 Non-orthogonal. 	$\text{ආ} = 0$  වෘත්ත 	ආ  $E_{\text{ආ}} \sim \text{රිම/Ⓜ වෘත්ත}$ Not detected by
රිම/ආ	 $\text{ආ} \sim \text{රිම/ආ වෘත්ත}$ 	 Non-orthogonal. $\text{ආ} = 0$ වෘත්ත 	ආ $\text{Ⓜ} \sim \text{රිම/ආ} \text{ } \square \square$
දිග	 දිග does not exist. 	 දිග does not exist. $\text{ආ} = 0$ වෘත්ත 	 Non-orthogonal.
Hall effect			
A	$\text{Ⓜ} \sim \text{රිම/ ආ වෘත්ත}$		
B	Ⓜ		
C	circuit. $\text{Ⓜ} \sim \text{රිම/ ආ වෘත්ත}$ is not in the measurement		
D	circuit. $\text{Ⓜ} \sim \text{රිම/ ආ වෘත්ත}$ is not in the measurement		
E	circuit. $\text{Ⓜ} \sim \text{රිම/ ආ වෘත්ත}$ is not in the measurement		

Table 2

Inverse	ଠ ଓଓଓଓ	ଠ ଓଓଓଓ	ଠ ଓଓଓଓ
ଓଓ	• Non-orthogonal.	• ଠ ଓଓଓଓ = 0 ଠ	• (ଠ ~ -ଓଓ/ଠ
ଓଓ/ଠ	• ଠ ଓଓ ~ - ଓଓ/ଠ ଓଓଓଓ • Not detected by inverse measurement.	• Non-orthogonal. • ଠ = 0 • ଓଓଓଓ	• ଠ • ଠ • ~ଓଓ/ଠ 0 ଠ 0 ଠ • Not detected by
ଓଓ	• (ଠ ~ ଓଓ ଓଓଓଓ	• ଠ = 0 • ଓଓଓଓ	• Non-orthogonal.
Region	Hall effect		
A	ଠ ~ -ଓଓ/ଠ ଓଓଓଓ		
B	ଠ		
C	ଠ ~ -ଓଓ/ଠ ଓଓଓଓ 0.	Yet no leading order contribution because $\int \text{ଓଓ/ଠ} = 0$.	
D	ଠ ~ -ଓଓ/ଠ ଓଓଓଓ 0.	Yet no leading order contribution because $\int \text{ଓଓ/ଠ} = 0$.	
E	ଠ ~ -ଓଓ/ଠ ଓଓଓଓ 0.	Yet no leading order contribution because $\int \text{ଓଓ/ଠ} = 0$.	

(5). COMSOL yields a z component of the stray field 0.083 T at point A, and a bypass current density $2.3 \times 10^8 \text{ A/m}^2$, resulting in a Hall electric field 0.01 A/m along the x direction (taking a Hall coefficient of Cu $R_H = 5.3 \times 10^{-10} \text{ m}^3 \text{ A}^{-1} \text{ s}^{-1}$). Considering a characteristic distance of 100 nm in the nanowire device, the Hall voltage is 1 nV. At point B, the x component of the stray field is - 0.14 T and the bypass current density is $1.0 \times 10^9 \text{ A/m}^2$, resulting in a Hall electric field 0.074 A/m along the z direction. Using the thickness 40 nm of the Cu nanowire, the Hall voltage is 3 nV. Hence, the Hall effect contribution is at the order of 1nV, which is approximately two orders of magnitude smaller than the experimentally measured signal (100 nV).

(6). By dividing the original nonlocal structure into n hypothetical finite volume elements, simulating the Hall current individually and adding up their contributions to the signal, we estimated the Hall voltage more precisely. The result was found to be less than 3 nV under the applied current of 500 μA . The

Hall effect-induced signal is then $\sim 0.006 \text{ m}\Omega$. The experimentally measured signal is $0.22 \text{ m}\Omega$, which is about two orders of magnitude larger than the calculated Hall effect contribution.

We added “The absolute value of the signal is two order larger than the artifact coming from the stray field-induced Hall effect in Cu estimated via the COMSOL simulation (Supplementary Section 3).” in Page 6, Line 114, “As the stray fields of the $\text{Co}_{25}\text{Fe}_{75}$ and the $\text{Ni}_{81}\text{Fe}_{19}$ are in the same order, the large difference between $|\Delta R|$ ($\text{Co}_{25}\text{Fe}_{75}$) and $|\Delta R|$ ($\text{Ni}_{81}\text{Fe}_{19}$) also excludes the stray field-induced artifacts in the nonlocal resistance.” in Page 6, Line 126 in the main text.

Furthermore, the Hall contribution shall not vanish at low temperature since the stray field and Hall effect exist. Yet the observed signal vanishes at low temperature, as shown in Fig. 4d. Hence, the Hall contribution cannot be the origin of the observed signal.

Fig.4 | Temperature dependence experiments. d, The temperature dependence results of $2\Delta R_{\text{DOEE}}^{(0)}$ (open diamonds with green color, left y-axis) and $2\Delta R_{\text{DOEE}}$ (closed circles and triangles with other colors, right y-axis). The error bars indicate the noise levels of R . The results are measured in samples with FM = $\text{Co}_{25}\text{Fe}_{75}$.

** Authors assume that the spin current generated when the current is injected through the FM plays a minimal role. I do not see any proof of that in their measurements, nor is the generated spin current negligible in all the temperature range measured. This could be easily verified through a non-local spin valve geometry by adding a second FM contact to the right side of the first FM. This should not be a problem for the authors, who are world-leading in the electrical characterization of such metallic spin valve devices.

Response: Following the reviewer's suggestion, we added a second FM contact to the right side of the first FM and measured the spin transport in room temperature. This structure allows us to measure the nonlocal spin valve (NLSV) and nonlocal inverse orbital Edelstein effect (IOEE) within the same device. For NLSV, both $\text{Ni}_{81}\text{Fe}_{19}$ and $\text{Co}_{25}\text{Fe}_{75}$ devices demonstrate the typical signals which increase as temperature decreases. For IOEE, only $\text{Co}_{25}\text{Fe}_{75}$ devices demonstrate the signals which decrease as temperature decreases, while the $\text{Ni}_{81}\text{Fe}_{19}$ devices show no significant signals at all temperatures. The results suggest that spin current plays a minor role in the observed signals. We also summarized the results in Supplementary Information section 5:

It is well established that charge-to-spin conversion is weak in systems composed of light elements due to the lack of strong spin-orbit coupling (SOC). Our device consists solely of Al_2O_3 and Cu where the charge-to-spin conversion is expected to be negligible. We designed an additional experiment to provide clear validation. Here, we fabricate the new device by introducing an additional FM nanowire into the original device, as illustrated in Fig. S8a. This modification allows us to simultaneously measure the nonlocal OEE response and the nonlocal spin injection response within a single device.

The conventional nonlocal spin valve (NLSV) measurement configuration is shown in Fig. S8a. In this setup, an electric current is applied through the lower FM nanowire, while the voltage signal V_{NLSV} is measured between the upper FM and the upper $\text{Al}_2\text{O}_3/\text{Cu}$ terminals. An external magnetic field is applied along the easy axis of FM. NLSV measurements were performed on two FM ($\text{Ni}_{81}\text{Fe}_{19}$ and $\text{Co}_{25}\text{Fe}_{75}$) at room temperature, with a separation distance of ~ 400 nm between the FMs and thickness of Cu 40nm. As shown in Fig. S8b and Fig. S8c, clear NLSV signals were observed in both $\text{Ni}_{81}\text{Fe}_{19}$ and $\text{Co}_{25}\text{Fe}_{75}$ devices, suggesting that spin current is injected into the $\text{Al}_2\text{O}_3/\text{Cu}$ nanowires.

The inverse orbital Edelstein effect measurement configuration is shown in Fig. S8d. In this setup, an electric current is applied through the lower FM nanowire, while the voltage signal V_{IOEE} is measured between the two ends of $\text{Al}_2\text{O}_3/\text{Cu}_y$ terminals. The distance between $\text{Al}_2\text{O}_3/\text{Cu}_y$ and lower FM nanowire is ~ 250 nm. However, for IOEE measurements, the two ferromagnetic materials exhibited starkly contrasting responses. In devices with $\text{FM} = \text{Ni}_{81}\text{Fe}_{19}$, no characteristic IOEE response was detected, while in $\text{Co}_{25}\text{Fe}_{75}$ devices, a clear IOEE signal was observed. This pronounced ferromagnetic material dependence of the IOEE signal differs significantly from conventional spin responses, which are typically less sensitive to the choice of ferromagnetic material. Based on these results, we attribute the observed signal, as shown in the main text, predominantly to an orbital response, while the contribution of spin injection is minimal.

Fig. S8 | Nonlocal spin valve measurement and inverse OEE measurement at room temperature. **a**, The NLSV measurement configuration. The spin current is injected into the $\text{Al}_2\text{O}_3/\text{Cu}$ nanowire. **b**, The NSVL measurement results of device with $\text{Ni}_{81}\text{Fe}_{19}$ at room temperature. **c**, The NSVL measurement results of device with $\text{Co}_{25}\text{Fe}_{75}$ at room temperature. **d**, The nonlocal IOEE measurement configuration. **e**, The nonlocal IOEE measurement of device with $\text{Ni}_{81}\text{Fe}_{19}$ at room temperature. No characteristic signal is observed. **f**, The nonlocal IOEE measurement of device with $\text{Co}_{25}\text{Fe}_{75}$ at room temperature. A typical nonlocal IOEE signal is observed.

We rewrote the sentence in main text Page 5, Line 101 as “Spin current is also generated in this process, but the negligible SOC in Cu^{16} should only lead to a tiny contribution to the voltage signal, as will be proved below.”,

and the sentence in Page 7, Line 129 as “The spin-induced nonlocal resistance also cannot strongly depend on the FM type (see Supplementary Section 5, where $\text{Co}_{25}\text{Fe}_{75}$ and $\text{Ni}_{81}\text{Fe}_{19}$ show comparable nonlocal spin valve signals), suggesting that the observed ΔR is dominated by OAM.”.

We also performed the temperature-dependent NLSV measurement as shown in Fig. 4c and 4e:

Fig. 4 | c, The typical signals for R_{NLSV} at 300 K (black squares) and 50 K (green circles). **e**, The summarized temperature dependence results of $2\Delta R_{\text{NLSV}}$ with $\text{Co}_{25}\text{Fe}_{75}$ (open blue squares) and $\text{Ni}_{81}\text{Fe}_{19}$ (open red circles). The error bars indicate the noise levels of R . The opposite temperature dependence behaviors of DOEE (d) and NLSV (e) implies distinct physics between OAM and SAM.

And we rewrote the sentence in Page 11, Line 209 as “Meanwhile, the nonlocal spin signal $2\Delta R_{\text{NLSV}}$ increases with decreasing temperature (Fig.4e) and presents in both devices with FM = $\text{Co}_{25}\text{Fe}_{75}$ and $\text{Ni}_{81}\text{Fe}_{19}$, clearly indicating the different physics involved in the observed nonlocal orbital and the spin signals”.

** Regarding Cu thickness-dependent experiments: Figure 3 shows an oxidized Cu layer between Al_2O_3 and non-oxidized Cu. According to the sample fabrication protocol described in the methods section, this oxidized Cu layer should be present in all the fabricated devices. However, when looking at Figure 1, this oxidized layer of Cu is not represented, and the authors state that the orbital angular momentum is created at the Al_2O_3 –Cu interface (caption of Figure 1). This is somewhat confusing that must be clarify, particularly because the authors are considering an interfacial effect. I believe the orbital angular momentum is created at the oxidized–non-oxidized Cu interface and not at the Al_2O_3 –Cu interface, which indeed does not exist in their

samples. Could the authors be more precise in explaining this? Also, how thick is the oxidized Cu layer formed by natural oxidation in air during 10 minutes?

Response: We thank the reviewer for highlighting the importance of accurately describing the sample structure and for raising valuable questions regarding the oxidized Cu layer. We modified the schematics in Fig. 1 and Fig. 3 of the main text:

Fig.1 | Schematic illustration of nonlocal transport measurements and verification of orbital response through ferromagnetic materials dependence experiments. a, Nonlocal measurement configuration to observe DOEE (direct measurement). The nonequilibrium OAM are generated by charge current I_c at the CuO_x/Cu interface through DOEE. The orbital accumulation then converts into SAM via SOC in FMs and induces a nonlocal response V . **b**, Nonlocal measurement configuration to observe IOEE (inverse measurement). A charge current brings nonequilibrium OAM from the FMs. The orbital accumulation then converts to charge current at the CuO_x/Cu interface through IOEE. In nonlocal measurement (**a** and **b**), the orbital generator and detector are sufficiently isolated in space with a separation distance d , allowing the measurement of nonlocal orbital response. **c**, **d**, Typical results of direct (**c**, R_{DOEE}) and inverse (**d**, R_{IOEE}) nonlocal orbital Edelstein resistance R which is defined as $R \equiv V/I_c$. The results are observed in sample A with separation distance $d = 140$ nm, Cu thickness $t_{\text{Cu}} = 40$ nm and FM = $\text{Co}_{25}\text{Fe}_{75}$ (Methods) while the external magnetic field B_{ext} swept along the hard axis of FM ($\Phi = 0^\circ$) from -1.25 T to 1.25 T. The signals are globally offset to position their center at $R = 0 \Omega$. The double-headed arrows indicate the definition of $2\Delta R_{\text{DOEE}}$ and

$2\Delta R_{\text{IOEE}}$, where $2\Delta R_{\text{DOEE}} = -2\Delta R_{\text{IOEE}} \approx 0.22 \text{ m}\Omega$. **e, f**, FM dependence results of $2\Delta R_{\text{DOEE}}$ (**e**) and $2\Delta R_{\text{IOEE}}$ (**f**). The solid curves represent the fitting of the data to Eq. (1), implying a long-range decay length of orbital accumulation λ_o of about 100 nm regardless of the selection of FMs. The dotted curves are guiding lines showing the value $R = 0 \Omega$. The error bars indicate the noise level of R . All results are in solid agreement with Onsager's reciprocal relations.

Fig.3 | Cu thickness dependence experiments. **a**, Schematic illustration of lateral and vertical OAM distribution and the role of oxidized and unoxidized Cu. The top Cu layer is oxidized and homogenous, assisting the long-range orbital response; the bottom Cu layer remains unoxidized, exhibiting a short vertical decay length of orbital accumulation. **b, c**, Cu thickness dependence results of $2\Delta R_{\text{DOEE}}$ (**b**) and $2\Delta R_{\text{IOEE}}$ (**c**). The solid curves represent the fitting of the data to Eq. (1). The error bars in (**b**) and (**c**) indicate the noise levels of R . **d**, λ_o at various Cu thicknesses obtained from fitting where the values are almost constant. The dotted line marks the value of $\lambda_o = 100 \text{ nm}$. **e**, Fitting result of parameter A . The solid curves show the exponential fitting for A . The vertical decay length of orbital accumulation λ_{z_o} ($\sim 25 \text{ nm}$) is much smaller than the lateral one λ_o ($\sim 100 \text{ nm}$), suggesting a distinction between lateral and vertical distribution of OAM. Error bars in (**d**) and (**e**) represent 95% confidence intervals from fitting results. All results agree with Onsager's reciprocal relations.

We also rewrote “In our samples, an oxidized Cu layer (denoted by CuO_x hereafter) which is expected to be about 3 nm thick (Supplementary Section 1), is sandwiched by the Al_2O_3 and Cu due to the natural oxidization (method).” in Page 5, Line 82, “In the direct measurement configuration (Fig. 1a), a charge current I_c is applied to the $\text{Al}_2\text{O}_3/\text{CuO}_x/\text{Cu}$ nanowire which is oriented along the y-axis in Fig. 1a (denoted by Cu_y nanowire hereafter).” in Page 5, Line 85, “A charge current I_c is injected into the $\text{Al}_2\text{O}_3/\text{CuO}_x/\text{Cu}$ nanowire which is oriented along the x-axis in Fig. 1b (denoted by Cu_x nanowire hereafter) from the FM magnetized along the x-axis.” in Page 5, Line 97.

To understand the thickness of oxidized Cu layer (CuO_x), we performed scanning transmission electron microscopy (STEM) and energy-dispersive X-ray spectrometry (EDX) on a thin film sample with Al_2O_3 (15 nm)/ CuO_x (exposed to air for 10 mins)/ Cu (40 nm). The thickness of the oxidized Cu layer is approximately 2~3 nm (STEM), which is shown in Supplementary Information section 1. Our results are consistent with previous research (Kim, J. *et al. Phys. Rev. Materials* **7**, L111401 (2023); Ding, S. *et al. Phys. Rev. Lett.* **128**, 067201 (2022); Ding, S. *et al. Phys Rev Res* **4**, L032041 (2022)).

Fig. S1 | STEM and EDX analysis results. **a**, STEM image for $\text{Al}_2\text{O}_3/\text{Cu}$ sample, representing the observed area for the EDX analysis. **b**, O, Al, Si, Cu atom EDX line profile near the $\text{Al}_2\text{O}_3/\text{Cu}$ interface.

Response to Reviewer #2:

The manuscript "Nonlocal Electrical Detection of Reciprocal Orbital Edelstein Effect" presents an experimental investigation of the Onsager reciprocity in orbital transport using nonlocal transport measurements. The authors use an $\text{Al}_2\text{O}_3/\text{Cu}/\text{ferromagnet}$ structure to test the reciprocal relation between direct and inverse orbital Edelstein effects (DOEE and IOEE). Their results demonstrate that reciprocal orbital responses produce equivalent electric voltages, supporting Onsager reciprocity. The authors also analyze the decay length of orbital accumulation with variations in Cu thickness and temperature, concluding that the behavior contrasts with spin transport dynamics.

This work addresses a largely unexplored aspect of orbitronics, specifically in nonlocal OAM transport, and the findings have the potential to advance the understanding of OAM dynamics. The experimental design is robust, and the analyses are well-detailed, though certain points could benefit from clarification and further examination.

Response: We appreciate the reviewer's insightful suggestions, which have significantly enhanced the quality and presentation of the manuscript. Our detailed responses are provided below.

1) Current Bypass Effect Explanation: The authors consider that the current bypass effect cannot account for the observed nonlocal response, as their simulations indicate a bypass effect decay length of 50 nm, compared to the experimental 100 nm. However, the difference between these two decay lengths may not be large enough to definitively rule out the bypass effect. It would strengthen the argument if the authors addressed the temperature dependence of the bypass effect to further separate it from the orbital accumulation. Such an analysis could clarify whether the observed temperature-dependent decay behavior is indeed related to the orbital effects or influenced by bypass effects.

Response: The bypass effect has nearly no temperature dependence. This is because the resistivity of FM is always one order larger than that of Cu, and the bypass current can be regarded as only flowing in the Cu layer. In a single material, as the resistivity is homogenous, the bypass current distribution is decided merely by the geometric of the device. We added the following discussion in Supplementary Section 10:

As discussed in Section 6, the diffusion length of the bypass current in our

device is approximately 47 nm at room temperature, while the decay length of orbital accumulation exceeds 100 nm. To conclusively exclude the influence of bypass current on our estimation of the orbital accumulation decay length, we performed low-temperature COMSOL simulations to assess the temperature dependence of the bypass current. Similarly, we simulated the surface

averaged value of $\frac{I_{by}}{1E7}$ across the cross section of the Cu_x nanowire above the

FM nanowires at various distances. The temperature dependent conductivities of Cu and FM (Fig. S17) were used as inputs to simulate the bypass current at various temperatures. The simulation result at 50 K is shown in Fig. S18a

suggesting an unchanged value of $\frac{I_{by}}{1E7}$ at various temperature, and the

diffusion length of bypass current A_{by} as a function of temperature are shown in Fig. S18b. A_{by} shows no temperature dependence, which is consistent with previous research, while A_o decreases at lower temperatures, suggesting that the observed temperature dependent decay behavior is primarily attributed to the orbital response and is not significantly influenced by the presence of bypass current.

Fig. S18 | Temperature dependence of bypass current. a, The bypass current as a function of distance under 50 K. Fitting the data to exponent decay equation (grey curve) gives a diffusion length of bypass current A_{by} about 47 nm, which is the same as the A_{by} at 300 K. **b,** The temperature dependence of the diffusion length bypass current. A_{by} remains 47 nm at various temperatures.

We then added “The temperature dependent λ_o also contrasts with the bypass effect which has no change at different temperature^{42,43} (supplementary section 10).” in Page 12, Line 222.

2) Mechanism of Long-Range Transport in Oxidized Cu: A critical element in the authors' interpretation is that the oxidized Cu layer enables long-range orbital transport due to its unique electronic band structure. This assumption is fundamental

to explaining the observed 100 nm decay length, but supporting evidence is limited. Providing further theoretical or empirical support for this mechanism would be helpful, as it is essential to distinguish the orbital response from other possible contributions. In fact, it could be beneficial to consider if a temperature-dependent mechanism like variable range hopping (VRH) might also influence transport in the oxidized Cu layer. VRH, often associated with transport in disordered systems, involves charge carriers hopping between localized states, a process that increases with temperature. If the transport in the oxidized Cu is dominated by VRH, the current through this layer could depend on temperature and, consequently, affect the OREE as well.

Examining whether VRH could be active in the oxidized Cu layer would be particularly relevant for interpreting the temperature dependence observed in the orbital accumulation. While it may not directly explain the differing decay lengths, VRH could provide insights into how temperature-dependent transport in the oxide influences overall current flow and orbital effects. If feasible, an analysis or discussion on the potential influence of VRH on orbital accumulation (or other mechanisms that the authors find relevant) would help clarify the role of the oxide layer in the observed temperature dependence.

Response: We thank the reviewer for the valuable suggestion. The idea of VRH is inspiring in that the conductivity of the oxidized Cu may influence the OAM propagation.

A fact is that an electron in a localized state cannot carry OAM by itself. Hence, in low temperature, the limited hopping between the localized states in the oxidized Cu cannot support orbital Rashba bands and long-range OAM propagation. The OAM propagation is more likely happening in the high temperature region, where free carriers can be excited due to the thermal broadening of the localized states, and a conductive channel is formed at the oxidized Cu layer. As a result, continuous orbital Rashba band is formed on the whole Cu nanowire, and long-range orbital propagation is allowed.

The OAM propagation is different from the typical VRH, where charge (which is always conserved) transport is considered. Also, our system contains Cu/oxidized Cu bilayer (metallic/semi-conductive), different from typical all-semi-conductive VRH systems. Hence, VRH theory might not be suitable for direct modelling of the OAM propagation in our system. We hereby consider a simple multiple-step hopping to describe the formation of continuous

conductive bands involving both Cu and CuO_x, i.e., the orbital Rashba bands, on the whole nanowire.

We added the following discussion as Supplementary Section 11:

We consider the hopping between the states in the oxidized Cu, which can be mediated by the metallic Cu, as illustrated in Fig. S19a. We assume that for each hopping between two neighboring grains, the probability for an electron to maintain its OAM information is

$$= C \exp(-\Delta E/k_B T) = \exp(-\alpha - \Delta E/k_B T), \quad (S11 - 1)$$

where $\alpha = -\ln C$ is a parameter related to the wavefunction overlapping, k_B is the Boltzmann constant, and ΔE is the level mismatch between the two neighboring grains. This ΔE is related to the inelastic processes that suppress the OAM. The OAM-maintaining hopping probability across N grains is

$$(N) = \exp(-N - N\Delta E/k_B T), \quad (S11 - 2)$$

and the total hopping distance is $d = Nr$, where r is the grain size. Thus, we can rewrite the hopping probability as

$$(N) = \exp[-d/\sigma] = \exp(-d/\sigma), \quad (S11 - 3)$$

where

$$\sigma = \frac{1}{\alpha} \frac{\Delta E}{k_B T} \quad (S11 - 4)$$

Taking $\Delta E / k_B = 10$ K, $r = 10$ nm, $\alpha = 1/15$, one can obtain a temperature dependence of λ_H (Fig. S19b), which is comparable with the experimental λ_H as shown in Fig. 5b.

Fig.S19 | a, Schematic of the multiple hopping in Cu/CuO_x bilayer. b, Temperature dependence of λ_H .

Fig. 5 | b, Fitting result of λ_0 for $2\Delta R_{\text{DOEE}}$ (blue) and $2\Delta R_{\text{IOEE}}$ (red) as functions of temperature. The error bars indicate the 95% confidence intervals from fitting results. All the results obey Onsager's reciprocal relations.

Such a hopping language description corresponds to that when conductive bands involving both Cu and oxidized Cu are formed continuously across distance λ_H , the eigenstates with OAM can be created and the OAM propagation is allowed across a similar distance.

We rewrote the sentences in Page 12, Line 226 as following:

“Therefore, the nonlocality may not be linked to a diffusive orbital current, but rather due to the eigenstates in the Rashba band which host OAM by themselves: the Rashba effect locks the nonequilibrium OAM locally to the diffusive linear momentum^{48,49} of the electrons. We propose a possible picture for the temperature dependence as shown in Fig. 5c. In low temperature, the limited hopping between the localized states in the oxidized Cu cannot support orbital Rashba bands. The itinerant electrons have wavefunctions constrained in the unoxidized Cu layer, and do not carry OAM. The OAM propagation is more likely happening in the high temperature region, where free carriers can be excited due to the thermal broadening of the localized states, and a conductive channel is formed at the oxidized Cu layer (Supplementary Section 11). The itinerant electrons then have wavefunctions penetrating into the oxidized Cu layer, creating hybridized states with OAM. Thus, continuous orbital Rashba bands are formed on the whole Cu nanowire, and long-range orbital propagation is allowed.”

And add a schematic as Fig.5c:

Fig.5 | Analysis on temperature dependent orbital transport. **a**, Fitting of the $2\Delta R_{\text{DOEE}}(T)$ data with Eq. (1). The error bars indicate the noise levels of R_{DOEE} . **b**, The fitting result of λ_0 for $2\Delta R_{\text{DOEE}}$ (blue) and $2\Delta R_{\text{IOEE}}$ (red) as functions of temperature. The error bars indicate the 95% confidence intervals from fitting results. All the results obey Onsager's reciprocal relations. **c**, Schematic of the proposed mechanism for the temperature-dependent orbital propagation. At low temperatures (upper panel), hopping between the localized states in the oxidized Cu (CuO_x layer) is limited, so the itinerant electrons are confined to the unoxidized Cu layer without carrying OAM ($L = 0$). Thus, continuous orbital Rashba bands are not forming. At high temperature (lower panel), the localized states are thermally broadened, allowing the conductive channel in the CuO_x layer. The electron wavefunctions are extended into the CuO_x layer, creating hybridized states with OAM ($L \neq 0$), so continuous orbital Rashba bands form and allow the long-range orbital propagation.

Response to Reviewer #3:

In this study, the author experimentally tested the Onsager reciprocity of orbital transport in an orbital Edelstein system by employing non-local measurements. This research provides a clear understanding of the intrinsic reciprocity relationship of orbital effects and reveals the long-range transverse correlation of orbital angular momentum accumulation, laying a foundation for the development of interconnected orbital electronics devices that can be operated over long distances. It's so interesting. However, there are some issues that need to be addressed before formal publication.

Response: We thank the reviewer for the nice evaluation. We modified our manuscript according to the reviewer's suggestions.

1. Although the spin Hall effect emphasizes that the material requires a large spin-orbit coupling strength, the spin-orbit coupling effect of the material itself can not be ignored. Thus, how can we determine whether the measurement result is due to the contribution of the orbital Hall effect, the spin Hall effect, or both?

Response: We thank the reviewer for the insightful question, which has helped us clarify this important aspect of our work. To address the reviewer's questions about distinguishing charge-spin and charge-orbital interconversions, we rely on the FM dependence of the observed signals. This approach is based on the fact that orbital angular momentum (OAM) must convert to spin angular momentum via spin-orbit coupling in the FM to generate a detectable potential difference.

As shown in Supplementary Information Section 5, to provide further evidence to distinguish charge-spin and charge-orbital interconversions, we perform an additional measurement of the spin current. The nonlocal spin valve (NLSV) measurement allows us to detect the spin current independently (Fig. S8a). An electric current is applied through the lower FM nanowire, inducing a spin current in the Cu nanowire which diffuses to the upper FM nanowire. A voltage signal V_{NLSV} is measured between the upper FM and the upper Cu terminals, corresponding to the spin chemical potential between Cu and FM. An external magnetic field is swapped along the easy axis of FM, aligning the magnetizations in the two FMs antiparallely at $\sim \pm 20$ mT due to the different coercivities (Fig. S8b). The difference at the parallel state ($B < 15$ mT and $B > 30$ mT) and the antiparallel state ($B \sim 20$ mT) corresponds to the chemical potential created by the nonequilibrium spin accumulation. As can be seen in Fig. S8b and c, both $\text{Co}_{25}\text{Fe}_{75}$ and $\text{Ni}_{81}\text{Fe}_{19}$ show clear NLSV signal, reflecting that spin current is generated in both materials.

We then performed nonlocal inverse orbital Edelstein effect (IOEE) measurements on the same devices. The electrical current is still applied from the lower FM nanowire, yet the voltage signal is measured by the $\text{Al}_2\text{O}_3/\text{Cu}$ nanowire (Fig. S8d). Here, $\text{Ni}_{81}\text{Fe}_{19}$ shows no signal (Fig. S8e) and only $\text{Co}_{25}\text{Fe}_{75}$ shows signal (Fig. S8f). Hence, the strong FM dependence of the IOEE signals further supports the conclusion that the detected signals are not dominated by spin currents but are instead driven by orbital responses. Thus, we attribute the measurement results to the orbital Edelstein effect rather than

the spin Hall effect.

Fig. S8 | Nonlocal spin valve measurement and inverse OEE measurement at room temperature. **a**, The NLSV measurement configuration. The spin current is injected into the Al₂O₃/Cu nanowire. **b**, The NLSV measurement results of device with Ni₈₁Fe₁₉ at room temperature. **c**, The NLSV measurement results of device with Co₂₅Fe₇₅ at room temperature. **d**, The nonlocal IOEE measurement configuration. **e**, The nonlocal IOEE measurement of device with Ni₈₁Fe₁₉ at room temperature. No characteristic signal is observed. **f**, The nonlocal IOEE measurement of device with Co₂₅Fe₇₅ at room temperature. A typical nonlocal IOEE signal is observed.

We rewrote the sentence in Page 5, Line 101 in main text as “Spin current is also generated in this process, but the negligible SOC in Cu¹⁶ should only lead to a tiny contribution to the voltage signal, as will be proved below”, and the sentence in Page 7, Line 129 as “The spin-induced nonlocal resistance also cannot strongly depend on the FM type (see Supplementary section 5, where Co₂₅Fe₇₅ and Ni₈₁Fe₁₉ show comparable nonlocal spin valve signals), suggesting that the observed ΔR is dominated by OAM”.

We also performed the temperature-dependent NLSV measurement as shown in Fig. 4c and 4e:

Fig. 4 c, Typical signals for R_{NLSV} at 300 K (black squares) and 50 K (green circles). **e**, The summarized temperature dependence results of $2\Delta R_{NLSV}$ with $Co_{25}Fe_{75}$ (open blue squares) and $Ni_{81}Fe_{19}$ (open red circles). The error bars indicate the noise levels of R . The opposite temperature dependent behaviors of DOEE (**d**) and NLSV (**e**) implies distinct physics between OAM and SAM.

It is clear that the spin current is enhanced when temperature decreases, in contrast with the voltage signal coming from the orbital-charge conversion, which shrinks when temperature reduces. Hence, we rewrote the sentence in Page 11, Line 209 as **“Meanwhile, the nonlocal spin signal $2\Delta R_{NLSV}$ increases with decreasing temperature (Fig.4e) and presents in both devices with FM = $Co_{25}Fe_{75}$ and $Ni_{81}Fe_{19}$, clearly indicating the different physics involved in the observed nonlocal orbital and the spin signals”**.

2. Why is there a strong dependence on ferromagnetic materials, suggesting that the measured signals originate from orbital responses rather than spin responses?

Response: The strong dependence on FM in the measured signals arises from the distinct mechanism of orbital torque, which differs fundamentally from spin torque. While the injection of a spin current into an FM exerts a torque on the magnetization through direct interaction with the local magnetic moment, orbital torque follows a different process. When an orbital current is injected into the FM, the OAM does not directly interact with the local magnetic moment. Instead, it must first be converted into spin angular momentum via spin-orbit coupling within the FM. The resulting spin angular momentum then interacts with the local magnetic moment through exchange coupling, generating the observed torque.

This additional orbital-to-spin conversion step is highly dependent on the spin-orbit coupling strength of the FM material, leading to the strong material dependence observed in the signals. This contrasts with conventional spin

torques, which are less sensitive to the FM material. As a result, the measured material dependence strongly suggests that the signals primarily originate from orbital responses rather than spin responses.

Moreover, as reported by Lee et al, $\text{Co}_{25}\text{Fe}_{75}$ [Fig. R1(a)] has a lot of hot spots near the Fermi surface hosting strong spin-orbit correlation. Less hot spots appear in $\text{Co}_{50}\text{Fe}_{50}$ [Fig. R1(b)]. This implies that $\text{Co}_{25}\text{Fe}_{75}$ can convert OAM into spin with higher efficiency, while such effect is weaker in $\text{Co}_{50}\text{Fe}_{50}$.

For $\text{Ni}_{81}\text{Fe}_{19}$ (approximated by $\text{Ni}_{75}\text{Fe}_{25}$) [Fig. R1(c)], the expected spin-orbit correlation is even weaker since no hot spot appears near the Fermi surface. These theoretical calculations align with our experimental results.

Fig. R1 | Band structures of (a) bcc $\text{Co}_{25}\text{Fe}_{75}$, (b) bcc $\text{Co}_{50}\text{Fe}_{50}$, (c) fcc $\text{Ni}_{75}\text{Fe}_{25}$. The horizontal dashed lines with $E = 0.0$ eV present the Fermi energy E_F . The unit of $\langle \mathbf{L} \cdot \mathbf{S} \rangle_{\text{FM}}$ (color) is \hbar^2 . (The figures are modified from Lee, H. & Lee, H.-W. *Current Applied Physics* 67, 60–68 (2024)).

3. Some pictures are too blurry, and the colors are too similar to distinguish. For example, figures 1(a) and 1(b), and figures 4(c) and 4(d).

Response: We thank the reviewer for pointing out the issue. We revised Fig. 1, Fig. 4 and Fig. 5 (containing the previous Fig. 4d) to enhance color contrast and improve data point distinguishability. The updated versions of Fig. 1, Fig. 4 and Fig. 5 are displayed below.

Fig.1 | Schematic illustration of nonlocal transport measurements and verification of orbital response through ferromagnetic materials dependence experiments. a, Nonlocal measurement configuration to observe DOEE (direct measurement). The nonequilibrium OAM are generated by charge current I_c at the CuO_x/Cu interface through DOEE. The orbital accumulation then converts into SAM via SOC in FMs and induces a nonlocal response V . **b,** Nonlocal measurement configuration to observe IOEE (inverse measurement). A charge current brings nonequilibrium OAM from the FMs. The orbital accumulation then converts to charge current at the CuO_x/Cu interface through IOEE. In nonlocal measurement (**a** and **b**), the orbital generator and detector are sufficiently isolated in space with a separation distance d , allowing the measurement of nonlocal orbital response. **c, d,** Typical results of direct (**c**, R_{DOEE}) and inverse (**d**, R_{IOEE}) nonlocal orbital Edelstein resistance R which is defined as $R \equiv V/I_c$. The results are observed in sample A with separation distance $d = 140$ nm, Cu thickness $t_{\text{Cu}} = 40$ nm and FM = $\text{Co}_{25}\text{Fe}_{75}$ (Methods) while the external magnetic field B_{ext} swept along the hard axis of FM ($\Phi = 0^\circ$) from -1.25 T to 1.25 T. The signals are globally offset to position their center at $R = 0$ Ω . The double-headed arrows indicate the definition of $2\Delta R_{\text{DOEE}}$ and $2\Delta R_{\text{IOEE}}$, where $2\Delta R_{\text{DOEE}} = -2\Delta R_{\text{IOEE}} \approx 0.22$ m Ω . **e, f,** FM dependence results

of $2\Delta R_{DOEE}$ (e) and $2\Delta R_{IOEE}$ (f). The solid curves represent the fitting of the data to Eq. (1), implying a long-range decay length of orbital accumulation λ_o of about 100 nm regardless of the selection of FMs. The dotted curves are guiding lines showing the value $R = 0 \Omega$. The error bars indicate the noise level of R . All results are in solid agreement with Onsager's reciprocal relations.

Fig.4 | Temperature dependence experiments. a, b, c, Typical signals for R_{DOEE} (a) $6\hat{\circ}\square EE$

(\square) (b) and R_{NLSV} (c) at 300 K (black squares) and 50 K (green circles). R_{DOEE} and $6\hat{\circ}\square EE$

(\square) signals decrease with lower temperatures, while R_{NLSV} signals increase as the temperature decreases. d, Temperature dependence results of $2\Delta 6\hat{\circ}\square EE$

(\square) (open diamonds with green color, left y-axis) and $2\Delta R_{DOEE}$ (closed circles and triangles with other colors, right y-axis). The error bars indicate the noise levels of R . The results are measured in samples with $FM = Co_{25}Fe_{75}$.

e, Summarized temperature dependence results of $2\Delta R_{NLSV}$ with $Co_{25}Fe_{75}$ (open blue squares) and $Ni_{81}Fe_{19}$ (open red circles). The error bars indicate the noise levels of R . The opposite temperature dependence on the behaviors of DOEE (d) and NLSV (e) implies distinct physics between OAM and SAM.

Fig.5 | Analysis on temperature dependent orbital transport. **a**, Fitting of the $2\Delta R_{\text{DOEE}}(T)$ data with Eq. (1). The error bars indicate the noise levels of R_{DOEE} . **b**, The fitting result of λ_0 for $2\Delta R_{\text{DOEE}}$ (blue) and $2\Delta R_{\text{IOEE}}$ (red) as functions of temperature. The error bars indicate the 95% confidence intervals from fitting results. All the results obey Onsager's reciprocal relations. **c**, Schematic of the proposed mechanism for the temperature-dependent orbital propagation. At low temperatures (upper panel), hopping between the localized states in the oxidized Cu (CuO_x layer) is limited, so the itinerant electrons are confined to the unoxidized Cu layer without carrying OAM ($L = 0$). Thus, continuous orbital Rashba bands are not forming. At high temperature (lower panel), the localized states are thermally broadened, allowing the conductive channel in the CuO_x layer. The electron wavefunctions are extended into the CuO_x layer, creating hybridized states with OAM ($L \neq 0$), so continuous orbital Rashba bands form and allow the long-range orbital propagation.

Response to Reviewer #1:

I have carefully read the authors' response to my concerns. While I am mostly convinced by their answers, there is still a critical issue regarding the CuO_x layer that I am not fully persuaded by.

Response: We thank the reviewer for the thorough evaluation. We provide detailed responses to each suggestion and comment below.

The formation of a 2 – 3 nm CuO_x layer -- which is a crucial ingredient for observing the Orbital Hall Edelstein Effect (OHEE)-- is supported by STEM and EDX measurements. However, I honestly do not clearly see how the authors conclude that this layer is precisely 2 – 3 nm thick. With the current EDX measurements, the resolution does not seem sufficient to make such a claim. Additionally, there is a non-negligible concentration of Al in the region where Cu and oxygen may form the oxide, which could further complicate the interpretation.

Response: We are sorry for the unclear description, and we agree that the thickness of the oxidized Cu layer cannot be precisely confirmed to be 2–3 nm. What can be concluded from the STEM and EDX measurements is that oxygen is detected within approximately 3 nm from the surface of the Cu layer, i.e., the CuO_x thickness is comparable to or less than 3 nm. This estimation should, therefore, be interpreted as an upper limit based on the available spatial resolution.

We rewrote the sentence on page 5 and line 82 of the main text: "In our samples, an oxidized Cu layer (denoted by CuO_x hereafter) < 3 nm thick (Supplementary Section 1), is formed in between the Al₂O₃ and Cu layers due to the natural oxidization (method)."

We rewrote the sentence on page 22, line 414 of methods in the main text: "The samples were exposed to the atmosphere at room temperature for 10 minutes before Al₂O₃ deposition. The Al₂O₃ capping layers to prevent Cu from further oxidation were only deposited on the Cu nanowires by electron beam deposition."

We also rewrote the sentences on page 3 and line 48 of the supplementary information: "According to the analysis, oxygen distributes with a concentration gradient confined within ~3 nm below the Cu surface, consistent with previously reported observations of Cu surface oxidation¹⁻³."

We acknowledge the presence of the Al element in the oxidized region. The Al_2O_3 is highly stable, and its Al–O bond dissociation energy ranges from 500 to 600 kJ/mol [Newman, R. N. et al. *Flame* 17, 149–157 (1971).; McCollum, J. Et al. *ACS Appl. Mater. Interfaces* 7, 18742–18749 (2015).; Li, G. et al. *Def. Technol.* 34, 313–327 (2024).]. Therefore, decomposing Al_2O_3 requires a much higher temperature than during our electron-beam deposition. Thus, we consider the presence of Al at the Cu surface to be due to the mixing of oxidized Al instead of metallic Al.

Based on previous experiments and reported theoretical studies, the presence of an Al element does not appear necessary for the emergence of the orbital Edelstein effect (OEE). In many reports, OEE has been observed in Cu with different oxide capping layers (Fig. R1) [Kim, J. et al. *Phys. Rev. B* 103, L020407 (2021).; Kim, J. et al. *Phys. Rev. Mater.* 7, L111401 (2023)] or without any intentional capping (natural oxidation) [Wang, H. et al. *Phys. Rev. Lett.* 134, 126701 (2025).; Ding, S., Wang, H., Legrand, W., Noël, P. & Gambardella, *Nano Lett.* 24, 10251–10257 (2024).; Ding, S. et al. *Phys. Rev. Lett.* 125, 177201 (2020)]. Additionally, theoretical calculations show that the interfacial oxidized Cu can exhibit a sizable OEE [Go, D. et al. *Phys. Rev. B* 103, L121113 (2021)]. These results demonstrated that when an oxide caps Cu, the surface oxidized Cu is the key factor responsible for OEE. Since the oxide dependence of the OEE is already well-established by these works, our current work focuses on the nonlocal response of OAM. It uses the demonstrated orbital Rashba system $\text{Al}_2\text{O}_3/\text{CuO}_x/\text{Cu}$ in all the primary samples.

Fig. R1 | The orbital torque efficiency of Cu capped by different oxide layers (modified from Kim, J. et al. *Phys. Rev. Mater.* 7, L111401 (2023)). Note that the orbital torque efficiency depends on the oxygen distribution ability of the oxide, i.e., how much the oxide can oxidize the Cu.

We had previously fabricated a sample consisting only of Cu and $\text{Co}_{25}\text{Fe}_{75}$ without any capping layer. In this device, when Cu was exposed to air at room temperature for about 24 hours, a clear nonlocal OEE signal was detected (Fig.

R2), supporting the idea that the observed orbital response does not depend on the Al element.

However, without a protective layer, the oxidation level of Cu can vary as a function of time, making a systematic comparison difficult. So, in our main set of experiments, we used an Al_2O_3 capping layer to control the oxidation level consistently for all samples.

Fig. R2 | The nonlocal measurements in Cu/Co₂₅Fe₇₅ sample (without Al₂O₃ capping layer). Direct measurement results in Cu/ Co₂₅Fe₇₅ (w/o Al₂O₃) sample, where the separation distance is 260 nm. This control device shows a reasonable DOEE signal at room temperature, suggesting that the observed orbital response does not depend on the Al element.

I believe there is a very simple yet effective experiment the authors should perform using a reference device, i.e., pure Cu capped with Al, where no Cu oxidation occurs. In this case, the observed OHEE, attributed to the CuO_x/Cu interface, should disappear and therefore not be detected. If the authors can perform this experiment and demonstrate that the effect is absent, I would be fully convinced of the validity of their observations.

Response: We thank the reviewer for the suggestion. As the reviewer proposed, a control experiment in which pure Cu is capped in situ immediately after deposition—thus preventing CuO_x formation—can be used to clarify whether Cu oxidation contributes to the observed OEE.

Considering experimental feasibility and our laboratory conditions, we selected Au as the capping layer. Au is highly stable at room temperature, effectively preventing Cu oxidation. With a full *d*-shell, Au is expected to exhibit a relatively small orbital Hall effect [Tanaka, T. *et al. Phys. Rev. B* 77, 165117 (2008).; Go, D., Lee, H.-W., Oppeneer, P. M., Blügel, S. & Mokrousov, Y. *Phys. Rev. B* 109, 174435 (2024)] and does not provide the *d* or *p* orbitals necessary to form an

OEE interface. Although a spin Rashba effect (charge-spin conversion) has been reported at the Au/Cu interface, the resulting signal in a nonlocal transport device is ~ 0.01 m Ω at room temperature [Pham, V. T. *et al. Phys. Rev. B* 104, 184410 (2021)], one order smaller than the room temperature signal we observed. Thus, we prepared new devices capped in situ with Au to prevent Cu oxidation, expecting no observable orbital response at room temperature.

Here, we fabricated devices consisting of 20 nm Co₂₅Fe₇₅, 40 nm Cu, and an in-situ 5 nm Au capping layer on the Cu. In room-temperature measurements, the Au/Cu sample (Fig. R3a, black squares) shows no signal, while the Al₂O₃/CuO_x/Cu sample shows a ~ 0.22 m Ω signal (Fig. R3b, black squares). Note that we amplified the signal in the Au/Cu sample 5 times in the plot.

At a low temperature (50 K), a small nonlocal signal (-0.02 m Ω) was detected in Au/Cu samples (Fig. R3a, green circles). The increased signal at low temperature demonstrates an opposite temperature dependence in the Cu/Au sample compared with that in the Al₂O₃/CuO_x/Cu sample.

Furthermore, comparing the 50 K signals for Au/Cu samples (Fig. R2a, green circles) to those for Al₂O₃/CuO_x/Cu samples (Fig. R2b), we observed that the signals have opposite signs, further supporting the different origins of these signals. We attribute the Au/Cu sample results to the spin Rashba effect (SRE) at the Au/Cu interface, which is consistent with previous reports [Pham, V. T. *et al. Phys. Rev. B* 104, 184410 (2021)]. Those findings demonstrated that the sizable signal from orbital transport diminished in a heterostructure without CuO_x and other sources with p and d orbitals for hybridization.

Fig. R3 | Nonlocal measurements in Au/Cu sample (a) and Al₂O₃/CuO_x/Cu sample (b). (a) The direct measurement results in an Au/Cu sample, where the separation distance is 300 nm. The typical nonlocal signal disappeared at room temperature, while a small signal caused by the direct spin Rashba effect

(DSRE) was detected at 50 K. The signals in the figure are magnified by 5 times and offset for clarity. The actual magnitude of OEE signals is 0.02 m Ω . **(b)** The direct measurement results in Al₂O₃/CuO_x/Cu samples, where the separating distance is 140 nm. The signals in the figure are offset for clarity.

We summarized the results in Supplementary Information section 10:

"As a control experiment, the samples of Au, Cu, and Co₂₅Fe₇₅ trilayers were measured. The control samples share the identical design as the experimental samples (Fig. 1a and 1b in main text). However, a 5 nm thick Au capping layer was in situ deposited by Joule heat deposition immediately after Cu deposition to prevent Cu oxidation. Au is highly stable at room temperature, effectively preventing Cu oxidation. With a full *d*-shell, Au provides no orbitals necessary for orbital hybridization, hence unlikely to support a strong orbital Rashba effect. Although a spin Rashba effect (SRE) has been reported at the Au/Cu interface, this effect is negligible at room temperature¹⁶.

The control experiment results are presented in Fig. S17. In the Au/Cu samples, no measurable nonlocal signal was observed at room temperature (Fig. S17a and Fig. S17b, black squares; Fig. S17c, open triangles), in contrast to the clear orbital response in Al₂O₃/CuO_x/Cu samples (Fig. 1c and 1d in the main text). At 50 K, a small signal (\sim 0.02 m Ω) appeared in the Au/Cu samples (Fig. S17a and Fig. S17b, green circles), whereas the signal was negligible in the Cu oxidized samples (Fig. 4a in main text). Moreover, the sign of the low-temperature signal in Au/Cu samples (green circles in Fig. S17a and Fig. S17b) is opposite to that in the Cu oxidized samples (black squares in Fig. 1c and 1d).

These three observations—the absence of a signal at room temperature, the distinct temperature dependence, and the opposite sign—demonstrate that the OEE is suppressed in the Au/Cu control devices. Instead, the signals at 50 K are likely due to the direct and inverse spin Rashba effect (DSRE and ISRE) at the Au/Cu interface, consistent with previous reports¹⁶. Those findings demonstrated that the sizable signal from orbital transport diminished in a heterostructure without CuO_x and other sources with *p* and *d* orbitals for hybridization.

Fig. S17 | The measurement results in Au/Cu devices. **a**, The results measured with direct measurement configuration. At room temperature, no signal was detected (black squares). At 50 K, a typical signal was observed, which is attributed to the direct spin Rashba effect (DSRE). **b**, The results measured with inverse measurement configuration. At room temperature, no signal was detected (black squares). At 50 K, a typical signal was observed due to the inverse spin Rashba effect (ISRE). **c**, The summarized data of control experiments. The error bars indicate the noise levels. Note that the signs of SRE signals in Au/Cu samples ($2\Delta R_{\text{DSRE}} < 0$, $2\Delta R_{\text{ISRE}} > 0$) are opposite to that of OEE signals in $\text{Al}_2\text{O}_3/\text{CuO}_x/\text{Cu}$ devices ($2\Delta R_{\text{DOEE}} > 0$, $2\Delta R_{\text{IOEE}} < 0$)."

We finally discuss other choices for the capping layer. If we deposit a metallic Al layer, due to its electron distribution $3s^23p^1$, p orbitals will appear near the Fermi surface in the system. Experiments have shown that Al can generate a sizable orbital torque [Rothschild, A. *et al. Phys. Rev. B* 106, 144415 (2022)], due to the sp hybridization. Meanwhile, a sizable orbital torque has also been reported in the Co/Al interface, attributed to the orbital Rashba effect [S. Krishnia *et al.*, *Nano Lett.* 23, 6785 (2023), S. A. Nikolaev *et al.*, *Nano Lett.* 24, 13465 (2024)]. Thus, metallic Al as a capping layer may still introduce orbital contributions and contaminate the measurement. Note that metallic Al and oxidized Al are Al elements. Still, they could have very different behaviors in forming the OEE interface since they provide different p orbital states for orbital hybridization.

Meanwhile, in-situ deposition of common oxide capping layers (e.g., Al_2O_3 , SiO_2 , TiO_2 , or MgO) can also cause Cu oxidation during the deposition process and establish an OEE interface [Kim, J. *et al. Phys. Rev. Mater.* 7, L111401 (2023)]. Hence, considering experimental feasibility and our laboratory conditions, we selected Au as the capping layer.

Response to Reviewer #2:

Following the revisions, the authors have significantly improved the manuscript by incorporating additional spin transport measurements, clarifying the roles of stray fields and bypass currents with extra COMSOL numerical calculations, and constructing a simplified theoretical model to explain the temperature dependence of the orbital response. These additions address the key concerns raised in the initial round of reports, strengthening the validity of their conclusions.

I now find the manuscript suitable for publication in its current form.

Response: We thank the reviewer for the careful review and kind support.

Response to Reviewer #3:

As I have mentioned in my first report, I consider the results that are presented in this manuscript very interesting and I believe that this result has the potential to grow into a new research direction in orbitronics and related materials. The concerns that I had for the first version of the manuscript have been answered in a satisfactory manner so I would recommend this manuscript to be published in Nature Communications.

Response: We thank the reviewer for the insightful suggestions, and the high evaluations.